# Where2comm: Communication-Efficient Collaborative Perception via Spatial Confidence Maps

**Yue Hu**         **Shaoheng Fang**         **Zixing Lei**
Cooperative Medianet Innovation Center, Shanghai Jiao Tong University
{18671129361, shfang, chezacarss}@sjtu.edu.cn

**Yiqi Zhong**                                    **Siheng  Chen**$^*$
University of Southern California     Shanghai Jiao Tong University, Shanghai AI Laboratory
yiqizhon@usc.edu                         sihengc@sjtu.edu.cn

## Abstract

Multi-agent collaborative perception could significantly upgrade the perception performance by enabling agents to share complementary information with each other through communication. It inevitably results in a fundamental trade-off between perception performance and communication bandwidth. To tackle this bottleneck issue, we propose a spatial confidence map, which reflects the spatial heterogeneity of perceptual information. It empowers agents to only share spatially sparse, yet perceptually critical information, contributing to where to communicate. Based on this novel spatial confidence map, we propose Where2comm, a communication-efficient collaborative perception framework. Where2comm has two distinct advantages: i) it considers pragmatic compression and uses less communication to achieve higher perception performance by focusing on perceptually critical areas; and ii) it can handle varying communication bandwidth by dynamically adjusting spatial areas involved in communication. To evaluate Where2comm, we consider 3D object detection in both real-world and simulation scenarios with two modalities (camera/LiDAR) and two agent types (cars/drones) on four datasets: OPV2V, V2X-Sim, DAIR-V2X, and our original CoPerception-UAVs. Where2comm consistently outperforms previous methods; for example, it achieves more than $100,000\times$ lower communication volume and still outperforms DiscoNet and V2X-ViT on OPV2V. Our code is available at https://github.com/MediaBrain-SJTU/where2comm.

## 1 Introduction

Collaborative perception enables multiple agents to share complementary perceptual information with each other, promoting more holistic perception. It provides a new direction to fundamentally overcome a number of inevitable limitations of single-agent perception, such as occlusion and long-range issues. Related methods and systems are desperately needed in a broad range of real-world applications, such as vehicle-to-everything-communication-aided autonomous driving [1–3], multi-robot warehouse automation system [4, 5] and multi-UAVs (unmanned aerial vehicles) for search and rescue [6–8]. To realize collaborative perception, recent works have contributed high-quality datasets [9–11] and effective collaboration methods [12, 13, 2, 14–19].

In this emerging field, the current biggest challenge is how to optimize the trade-off between perception performance and communication bandwidth. Communication systems in real-world scenarios are always constrained that they can hardly afford huge communication consumption in real-time, such as passing complete raw observations or a large volume of features. Therefore,

---

$^*$Corresponding author

36th Conference on Neural Information Processing Systems (NeurIPS 2022).

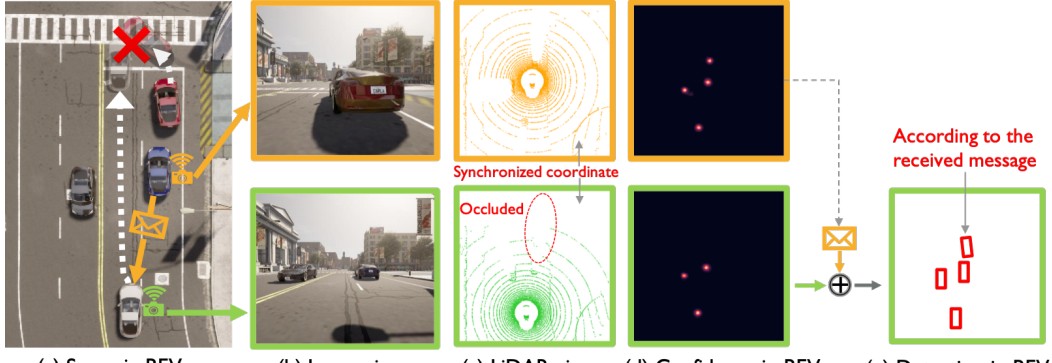

| (a) Scene in BEV | (b) Image view | (c) LiDAR view | (d) Confidence in BEV | (e) Detection in BEV |

Figure 1: Collaborative perception could contribute to safety-critical scenarios, where the white car and the red car may collide due to occlusion. This collision could be avoided when the blue car can share a message about the red car's position. Such a message is spatially sparse, yet perceptually critical. Considering the precious communication bandwidth, each agent needs to speak to the point!

we cannot solely promote the perception performance without evaluating the expense of every bit of precious communication bandwidth. To achieve a better performance and bandwidth trade-off, previous works put forth solutions from several perspectives. For example, When2com [12] considers a handshake mechanism which selects the most relevant collaborators; V2VNet [1] considers end-to-end-learning-based source coding; and DiscoNet [2] uses 1D convolution to compress message. However, all previous works make a plausible assumption: once two agents collaborate, they are obligated to share perceptual information of all spatial areas *equally*. This unnecessary assumption can hugely waste the bandwidth as a large proportion of spatial areas may contain irrelevant information for perception task. Figure 1 illustrates such a spatial heterogeneity of perceptual information.

To fill this gap, we consider a novel spatial-confidence-aware communication strategy. The core idea is to enable a spatial confidence map for each agent, where each element reflects the perceptually critical level of a corresponding spatial area. Based on this map, agents decide which spatial area (where) to communicate about. That is, each agent offers spatially sparse, yet critical features to support other agents, and meanwhile requests complementary information from others through multi-round communication to perform efficient and mutually beneficial collaboration.

Following this strategy, we propose `Where2comm`, a novel communication-efficient multi-agent collaborative perception framework with the guidance of spatial confidence maps; see Fig. 2. `Where2comm` includes three key modules: i) a spatial confidence generator, which produces a spatial confidence map to indicate perceptually critical areas; ii) a spatial confidence-aware communication module, which leverages the spatial confidence map to decide *where* to communicate via novel message packing, and *who* to communicate via novel communication graph construction; and iii) a spatial confidence-aware message fusion module, which uses novel confidence-aware multi-head attention to fuse all messages received from other agents, upgrading the feature map for each agent.

`Where2comm` has two distinct advantages. First, it promotes pragmatic compression at the feature level and uses less communication to achieve higher perception performance by focusing on perceptually critical areas. Second, it adapts to various communication bandwidths and communication rounds, while previous models only handle one predefined communication bandwidth and a fixed number of communication rounds. To evaluate `Where2comm`, we consider the collaborative 3D object detection task on four datasets: DAIR-V2X [11], V2X-Sim [9], OPV2V [10] and our original dataset CoPerception-UAVs. Our experiments cover both real-world and simulation scenarios, two types of agents (cars and drones) and sensors (LiDAR and cameras). Results show that i) the proposed `Where2comm` consistently and significantly outperforms previous works in the performance-bandwidth trade-off across multiple datasets and modalities; and ii) `Where2comm` achieves better trade-off when the communication round increases.

## 2 Related Works

**Multi-agent communication.** The communication strategy in multi-agent systems has been widely studied [20]. Early works [21–23] often use predefined protocols or heuristics to decide how agents communicate with each other. However, it is difficult to generalize those methods to complex tasks. Recent works, thus, explore learning-based methods for complex scenarios. For example,

Table 1: Major components comparisons of collaborative perception systems.

| Method | Venue | Message packing | Communication graph construction | Message fusion |
|---|---|---|---|---|
| When2com [12] | CVPR 2020 | Full feature map | Handshake-based sparse graph | Attention per-agent |
| V2VNet [1] | ECCV 2020 | Full feature map | Fully connected graph | Average per-agent |
| DiscoNet [2] | NeurIPS 2021 | Full feature map | Fully connected graph | MLP-based attention per-location |
| V2X-ViT [26] | ECCV 2022 | Full feature map | Fully connected graph | Self-attention per-location |
| Where2comm | NeurIPS 2022 | *Confidence*-aware sparse feature map + request map | *Confidence*-aware sparse graph | *Confidence*-aware multi-head attention per-location |

CommNet [24] learns continuous communication in the multi-agent system. Vain [25] adopts the attention mechanism to help agents selectively fuse the information from others. Most of these previous works consider decision-making tasks and adopt reinforcement learning due to the lack of explicit supervision. In this work, we focus on the perception task. Based on direct perception supervision, we apply supervised learning to optimize the communication strategy in both trade-off perception ability and communication cost.

**Collaborative perception.** As a recent application of multi-agent communication systems to perception tasks, collaborative perception is still immature. To support this area of research, there is a surge of high-quality datasets (e.g., V2X-Sim [9], OpenV2V [10], Comap[27] and DAIR-V2X[11]), as well as collaboration methods aimed for better performance-bandwidth trade-off (see comparisons in Table 1). When2com [12] proposes a handshake communication mechanism to decide *when* to communicate and create sparse communication graph. V2VNet [1] proposes multi-round message passing based on graph neural networks to achieve better perception and prediction performance. DiscoNet [2] adopts knowledge distillation to take the advantage of both early and intermediate collaboration. OPV2V [10] proposes a graph-based attentive intermediate fusion to improve perception performances. V2X-ViT [26] introduces a novel heterogeneous multi-agent attention module to fuse information across heterogeneous agents. In this work, we leverage the proposed spatial confidence map to promote more compact messages, more sparse communication graphs, and more comprehensive fusion, resulting in efficient and effective collaboration.

## 3 Problem Formulation

Consider $N$ agents in the scene. Let $\mathcal{X}_i$ and $\mathcal{Y}_i$ be the observation and the perception supervision of the $i$th agent, respectively. The objective of collaborative perception is to achieve the maximized perception performance of all agents as a function of the total communication budge $B$ and communication round $K$; that is,

$$\xi_\Phi(B,K) = \arg\max_{\theta,\mathcal{P}} \sum_{i=1}^{N} g\left(\Phi_\theta\left(\mathcal{X}_i, \{\mathcal{P}_{i \to j}^{(K)}\}_{j=1}^{N}\right), \mathcal{Y}_i\right), \ \text{s.t.} \sum_{k=1}^{K}\sum_{i=1}^{N} |\mathcal{P}_{i \to j}^{(k)}| \le B,$$

where $g(\cdot,\cdot)$ is the perception evaluation metric, $\Phi$ is the perception network with trainable parameter $\theta$, and $\mathcal{P}_{i \to j}^{(k)}$ is the message transmitted from the $i$th agent to the $j$th agent at the $k$th communication round. Note that i) when $B = K = 0$, there is no collaboration and $\xi_\Phi(0,0)$ reflects the single-agent perception performance; ii) through optimizing the communication strategy and the network parameter, collaborative perception should perform well consistently at any communication bandwidth or round; and iii) we consider multi-round communication, where each agent serves as both a supporter (offering message to help others) and a requester (requesting messages from others).

In this work, we consider the perception task of 3D object detection and present three contributions: i) we make communication more efficient by designing compact messages and sparse communication graphs; ii) we boost the perception performance by implementing more comprehensive message fusion; iii) we enable the overall system to adapt to varying communication conditions by dynamically adjusting where and who to communicate.

## 4 `Where2comm`: Spatial Confidence-Aware Collaborative Perception System

This section presents `Where2comm`, a multi-round, multi-modality, multi-agent collaborative perception framework based on a spatial-confidence-aware communication strategy; see the overview in Fig. 2. `Where2comm` includes an observation encoder, a spatial confidence generator, the spatial confidence-aware communication module, the spatial confidence-aware message fusion module and a detection decoder. Among five modules, the proposed spatial confidence generator generates the spatial confidence map. Based on this spatial confidence map, the proposed spatial confidence-aware communication generates compact messages and sparse communication graphs to save communication bandwidth; and the proposed spatial confidence-aware message fusion module leverages

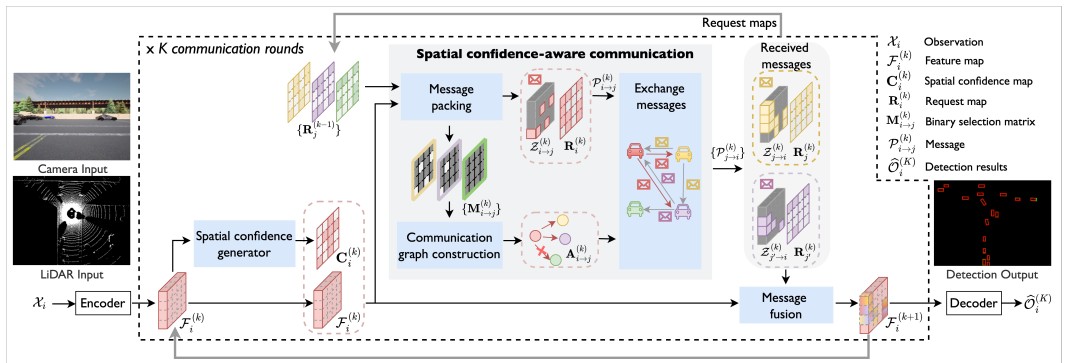

Figure 2: System overview. In `Where2comm`, spatial confidence generator enables the awareness of spatial heterogeneous of perceptual information, spatial confidence-aware communication enables efficient communication, and spatial confidence-aware message fusion boosts the performance.

informative spatial confidence priors to achieve better aggregation; also see an algorithmic summary in Algorithm 1 and the optimization-oriented design rationale in Section 7.3 in Appendix.

## 4.1 Observation encoder

The observation encoder extracts feature maps from the sensor data. `Where2comm` accepts single/multi-modality inputs, such as RGB images and 3D point clouds. This work adopts the feature representations in bird's eye view (BEV), where all agents project their individual perceptual information to the same global coordinate system, avoiding complex coordinate transformations and supporting better shared cross-agent collaboration. For the $i$th agent, given its input $\mathcal{X}_i$, the feature map is $\mathcal{F}_i^{(0)} = \Phi_{\mathrm{enc}}(\mathcal{X}_i) \in \mathbb{R}^{H \times W \times D}$, where $\Phi_{\mathrm{enc}}(\cdot)$ is the encoder, the superscript 0 reflects that the feature is obtained before communication and $H, W, D$ are its height, weight and channel. All agents share the same BEV coordinate system. For the image input, $\Phi_{\mathrm{enc}}(\cdot)$ is followed by a warping function that transforms the extracted feature from front-view to BEV. For 3D point cloud input, we discretize 3D points as a BEV map and $\Phi_{\mathrm{enc}}(\cdot)$ extracts features in BEV. The extracted feature map is output to the spatial confidence generator and the message fusion module.

## 4.2 Spatial confidence generator

The spatial confidence generator generates a spatial confidence map from the feature map of each agent. The spatial confidence map reflects the perceptually critical level of various spatial areas. Intuitively, for object detection task, the areas that contain objects are more critical than background areas. During collaboration, areas with objects could help recover the miss-detected objects due to the limited view; and background areas could be omitted to save the precious bandwidth. So we represent the spatial confidence map with the detection confidence map, where the area with high perceptually critical level is the area that contains an object with a high confidence score.

To implement, we use a detection decoder structure to produce the detection confidence map. Given the feature map at the $k$th communication round, $\mathcal{F}_i^{(k)}$, the corresponding spatial confidence map is

$$\mathbf{C}_i^{(k)} = \Phi_{\mathrm{generator}}(\mathcal{F}_i^{(k)}) \in [0, 1]^{H \times W}, \tag{1}$$

where the generator $\Phi_{\mathrm{generator}}(\cdot)$ follows a detection decoder. Since we consider multi-round collaboration, `Where2comm` iteratively updates the feature map by aggregating information from other agents. Once $\mathcal{F}_i^{(k)}$ is obtained, (1) is triggered to reflect the perceptually critical level at each spatial location. The proposed spatial confidence map answers a crucial question that was ignored by previous works: for each agent, information at which spatial area is worth sharing with others. By answering this, it provides a solid base for efficient communication and effective message fusion.

## 4.3 Spatial confidence-aware communication

With the guidance of spatial confidence maps, the proposed communication module packs compact messages with spatially sparse feature maps and transmits messages through a sparsely-connected communication graph. Most existing collaboration perception systems [1, 2, 26] considers full feature maps in the messages and fully-connected communication graphs. To reduce the communication bandwidth without affecting perception, we leverage the spatial confidence map to select the most

informative spatial areas in the feature map (where to communicate) and decide the most beneficial collaboration partners (who to communicate).

**Message packing.** Message packing determines what information should be included in the to-be-sent message. The proposed message includes: i) a request map that indicates at which spatial areas the agent needs to know more; and ii) a spatially sparse, yet perceptually critical feature map.

The request map of the $i$th agent is $\mathbf{R}_i^{(k)} = 1 - \mathbf{C}_i^{(k)} \in \mathbb{R}^{H \times W}$, negatively correlated with the spatial confidence map. The intuition is, for the locations with low confidence score, an agent is hard to tell if there is really no objects or it is just caused by the limited information (e.g. occlusion). Thus, the low confidence score indicates there could be missing information at that location. Requesting information at these locations from other agents could improve the current agent's detection accuracy.

The spatially sparse feature map are selected based on each agent's spatial confidence map and the received request maps from others. Specifically, a binary selection matrix is used to represent each location is selected or not, where 1 denotes selected, and 0 elsewhere. For the message sent from the $i$th agent to the $j$th agent at the $k$th communication round, the binary selection matrix is

$$\mathbf{M}_{i \to j}^{(k)} = \begin{cases} \Phi_{\text{select}}(\mathbf{C}_i^{(k)}) \in \{0,1\}^{H \times W}, & k = 0; \\ \Phi_{\text{select}}(\mathbf{C}_i^{(k)} \odot \mathbf{R}_j^{(k-1)}), \in \{0,1\}^{H \times W}, & k > 0; \end{cases} \quad (2)$$

where $\odot$ is the element-wise multiplication, $\mathbf{R}_j^{(k-1)}$ is the request map from the $j$th agent received at the previous round, $\Phi_{\text{select}}(\cdot)$ is the selection function which targets to select the most critical areas conditioned on the input matrix, which represents the critical level at the certain spatial location. We implement $\Phi_{\text{select}}(\cdot)$ by selecting the locations where the largest elements at in the given input matrix conditioned on the bandwidth limit; optionally, a Gaussian filter could be applied to filter out the outliers and introduce some context. In the initial communication round, each agent selects the most critical areas from its own perspective as the request maps from other agents are not available yet; in the subsequent rounds, each agent also takes the partner's request into account, enabling more targeted communication. Then, the selected feature map is obtained as $\mathcal{Z}_{i \to j}^{(k)} = \mathbf{M}_{i \to j}^{(k)} \odot \mathcal{F}_i^{(k)} \in \mathbb{R}^{H \times W \times D}$, which provides spatially sparse, yet perceptually critical information.

Overall, the message sent from the $i$th agent to the $j$th agent at the $k$th communication round is $\mathcal{P}_{i \to j}^{(k)} = (\mathbf{R}_i^{(k)}, \mathcal{Z}_{i \to j}^{(k)})$. Note that i) $\mathbf{R}_i^{(k)}$ provides spatial priors to request complementary information for the $i$th agent's need in the next round; the feature map $\mathcal{Z}_{i \to j}^{(k)}$ provides supportive information for the $i$th agent's need in the this round. They together enable mutually beneficial collaboration; ii) since $\mathcal{Z}_{i \to j}^{(k)}$ is sparse, we only transmit non-zero features and corresponding indices, leading to low communication cost; and iii) the sparsity of $\mathcal{Z}_{i \to j}^{(k)}$ is determined by the binary selection matrix, which dynamically allocates the communication budget at various spatial areas based on their perceptual critical level, adapting to various communication conditions.

**Communication graph construction.** Communication graph construction targets to identify when and who to communicate to avoid unnecessary communication that wastes the bandwidth. Most previous works [1, 2, 10] consider fully-connected communication graphs. When2com [12] proposes a handshake mechanism, which uses similar global features to match partners. This is hard to interpret because two agents, which have similar global features, do not necessarily need information from each other. Different from all previous works, we provide an explicit design rationale: the necessity of communication between the $i$th and the $j$th agents is simply measured by the overlap between the information that the $i$th agent has and the information that the $j$th agent needs. With the help of the spatial confidence map and the request map, we construct a more interpretable communication graph.

For the initial communication round, every agent in the system is not aware of other agents yet. To activate the collaboration, we construct a fully-connected communication graph. Every agent will broadcast its message to the rest of the system. For the subsequent communication rounds, we examine if the communication between agent $i$ and agent $j$ is necessary based on the maximum value of the binary selection matrix $\mathbf{M}_{i \to j}^{(k)}$, i.e. if there is at least one patch is activated, then we regard the connection is necessary. Formally, let $\mathbf{A}^{(k)}$ be the adjacency matrix of the communication graph at the $k$th communication round, whose $(i,j)$th element is

$$\mathbf{A}_{i,j}^{(k)} = \begin{cases} 1, & k = 0; \\ \max_{h \in \{0,1,..,H-1\}, w \in \{0,1,...,W-1\}} \left(\mathbf{M}_{i \to j}^{(k)}\right)_{h,w} \in \{0,1\}, & k > 0; \end{cases}$$

where $h, w$ index the spatial area, reflecting message passing from the $i$th agent to the $j$th agent. Given this sparse communication graph, agents can exchange messages with selected partners.

## 4.4 Spatial confidence-aware message fusion

Spatial confidence-aware message fusion targets to augment the feature of each agent by aggregating the received messages from the other agents. To achieve this, we adopt a transformer architecture, which leverages multi-head attention to fuse the corresponding features from multiple agents at each individual spatial location. The key technical design is to include the spatial confidence maps of all the agents to promote cross-agent attention learning. The intuition is that, the spatial confidence map could explicitly reflect the perceptually critical level, providing a useful prior for attention learning.

Specifically, for the $i$th agent, after receiving the $j$th agent's message $\mathcal{P}_{j \to i}^{(k)}$, it could unpack to retrieve the feature map $\mathcal{Z}_{j \to i}^{(k)}$ and the spatial confidence map $\mathbf{C}_j^{(k)} = 1 - \mathbf{R}_j^{(k)}$. We also include the ego feature map in fusion and denote $\mathcal{Z}_{i \to i}^{(k)} = \mathcal{F}_i^{(k)}$ to make the formulation simple and consistent, where $\mathcal{Z}_{i \to i}^{(k)}$ might not be sparse. To fuse the features from the $j$th agent at the $k$th communication round, the cross-agent/ego attention weight for the $i$th agent is

$$\mathbf{W}_{j \to i}^{(k)} = \text{MHA}_{\text{W}}\left(\mathcal{F}_i^{(k)}, \mathcal{Z}_{j \to i}^{(k)}, \mathcal{Z}_{j \to i}^{(k)}\right) \odot \mathbf{C}_j^{(k)} \in \mathbb{R}^{H \times W}, \tag{3}$$

where $\text{MHA}_{\text{W}}(\cdot)$ is a multi-head attention applied at each individual spatial location, which outputs the scaled dot-product attention weight. Note that i) the proposed spatial confidence maps contributes to the attention weight, as the features with higher perceptually critical level are more preferred in the feature aggregation; ii) the cross-agent attention weight models the collaboration strength with a $H \times W$ spatial resolution, leading to more flexible information fusion at various spatial regions. Then, the feature map of the $i$th agent after fusing the messages in the $k$th communication round is

$$\mathcal{F}_i^{(k+1)} = \text{FFN}\left(\sum_{j \in \mathcal{N}_i \bigcup \{i\}} \mathbf{W}_{j \to i}^{(k)} \odot \mathcal{Z}_{j \to i}^{(k)}\right) \in \mathbb{R}^{H \times W \times D},$$

where $\text{FFN}(\cdot)$ is the feed-forward network and $\mathcal{N}_i$ is the neighbors of the $i$th agent defined in the communication graph $\mathbf{A}^{(k)}$. The fused feature $\mathcal{F}_i^{(k+1)}$ would serve as the $i$th agent's feature in the $(k+1)$th round. In the final round, we output $\mathcal{F}_i^{(k+1)}$ to the detection decoder to generate detections.

**Sensor positional encoding.** Sensor positional encoding represents the physical distance between each agent's sensor and its observation. It adopts a standard positional encoding function conditioned on the sensing distance and feature dimension. The features are summed up with the positional encoding of each location before inputting to the transformer.

Compared to existing fusion modules that do not use attention mechanism [1] or only use agent-level attentions [12], the per-location attention mechanism adopted by the proposed fusion emphasizes the location-specific feature interactions. It makes the feature fusion more targeted. Compared to the methods that also use the per-location attention-based fusion module[2, 10, 26], the proposed fusion module leverages multi-head attention with two extra priors, including spatial confidence map and sensing distances. Both assist attention learning to prefer high quality and critical features.

## 4.5 Detection decoder

The detection decoder decodes features into objects, including class and regression output. Given the feature map at the $k$th communication round $\mathcal{F}_i^{(k)}$, the detection decoder $\Phi_{\text{dec}}(\cdot)$ generate the detections of $i$th agent by $\widehat{\mathcal{O}}_i^{(k)} = \Phi_{\text{dec}}(\mathcal{F}_i^{(k)}) \in \mathbb{R}^{H \times W \times 7}$, where each location of $\widehat{\mathcal{O}}_i^{(k)}$ represents a rotated box with class $(c, x, y, h, w, \cos \alpha, \sin \alpha)$, denoting class confidence, position, size and angle. The objects are the final output of the proposed collaborative perception system. Note that $\widehat{\mathcal{O}}_i^{(0)}$ denotes the detections without collaboration.

## 4.6 Training details and loss functions

To train the overall system, we supervise two tasks: spatial confidence generation and object detection at each round. As mentioned before, the functionality of the spatial confidence generator is the same as the classification in the detection decoder. To promote parameter efficiency, our spatial confidence generator reuses the parameters of the detection decoder. For the multi-round settings, each round is

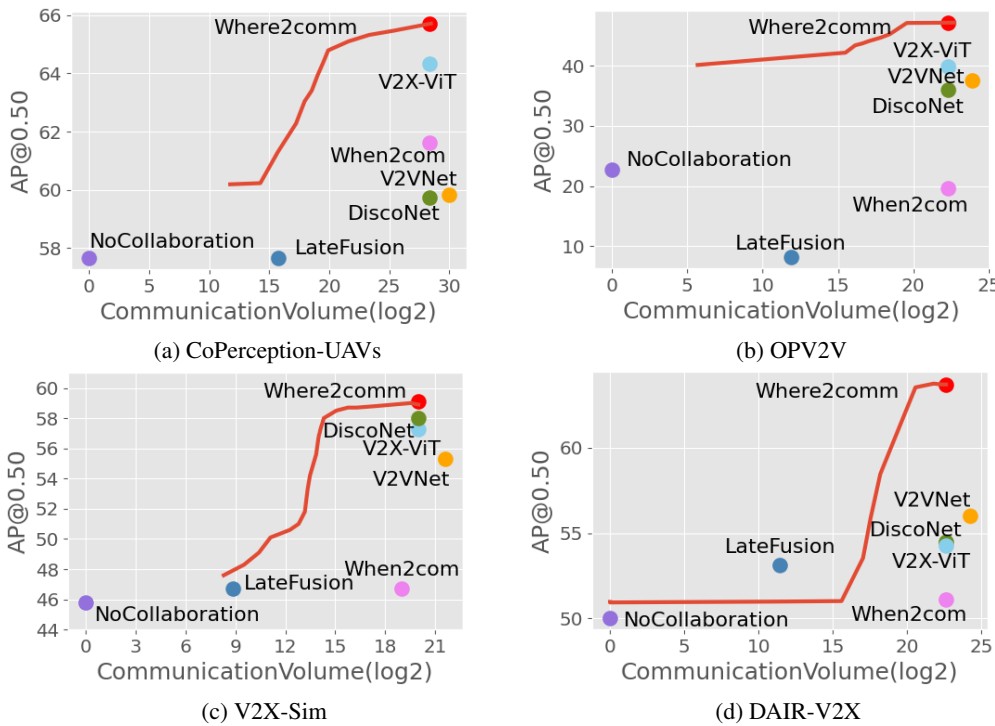

(a) CoPerception-UAVs

(b) OPV2V

(c) V2X-Sim

(d) DAIR-V2X

Figure 3: `Where2comm` achieves consistently superior performance-bandwidth trade-off on all the three collaborative perception datasets, e.g, `Where2comm` achieves *5,000* times less communication volume and still outperforms When2com on CoPerception-UAVs dataset. The entire red curve comes from a single `Where2comm` model evaluated at varying bandwidths.

supervised with one detection loss, the overall loss is $L = \sum_{k=0}^{K} \sum_{i}^{N} L_{\text{det}}\left(\widehat{\mathcal{O}}_i^{(k)}, \mathcal{O}_i\right)$, where $\mathcal{O}_i$ is the $i$th agent's ground-truth objects, $L_{\text{det}}$ is the detection loss [28].

**Training strategy for multi-round setting.** To adapt to multi-round communication and dynamic bandwidth, we train the model under various communication settings with curriculum learning strategy [29]. We first gradually increase the communication bandwidth and round; and then, randomly sample bandwidth and round to promote robustness. Through this training strategy, a single model can perform well at various communication conditions.

## 5 Experimental Results

Our experiments covers four datasets, both real-world and simulation scenarios, two types of agents (cars and drones) and two types of sensors (LiDAR and cameras). Specifically, we conduct camera-only 3D object detection in the setting of V2X-communication aided autonomous driving on OPV2V dataset [10], camera-only 3D object detection in the setting of drone swarm on the proposed CoPerception-UAVs dataset, and LiDAR-based 3D object detection on DAIR-V2X dataset [11] and V2X-Sim dataset [9]. The detection results are evaluated by Average Precision (AP) at Intersection-over-Union (IoU) threshold of $0.50$ and $0.70$. The communication results count the message size by byte in log scale with base $2$. To compare communication results straightforward and fair, we do not consider any extra data/feature/model compression.

### 5.1 Datasets and experimental settings

**OPV2V.** OPV2V [10] is a vehicle-to-vehicle collaborative perception dataset, co-simulated by OpenCDA [10] and Carla [30]. It includes 12K frames of 3D point clouds and RGB images with 230K annotated 3D boxes. The perception range is 40m×40m. For camera-only 3D object detection task on OPV2V, we implement the detector following CADDN [31]. The input front-view image size is $(416, 160)$. The front-view input feature map is transformed to BEV with resolution 0.5m/pixel.

**V2X-Sim.** V2X-Sim [9] is a vehicle-to-everything collaborative perception dataset, co-simulated by SUMO [32] and Carla, including 10K frames of 3D LiDAR point clouds and 501K 3D boxes.

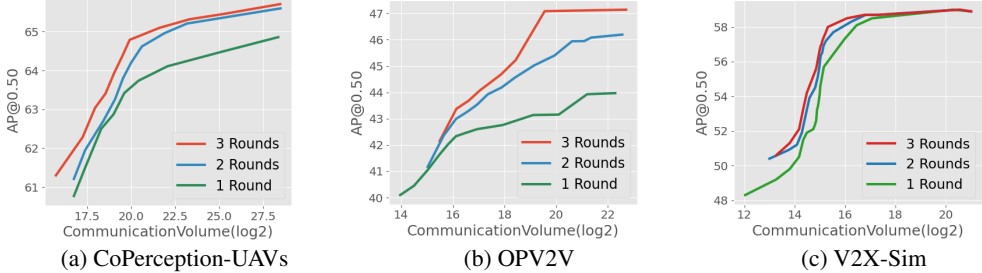

| (a) CoPerception-UAVs | (b) OPV2V | (c) V2X-Sim |

Figure 4: More communication rounds continuously improve performance-bandwidth trade-off.

The perception range is 64m×64m. For LiDAR-based 3D object detection task, our detector follows MotionNet [33]. We discretize 3D points into a BEV map with size $(256, 256, 13)$ and the resolution is $0.4$m/pixel in length and width, $0.25$m in height.

**CoPerception-UAVs.** To enrich the collaborative perception datasets, we consider the swarm of unmanned aerial vehicles (UAV) and propose a UAV-swarm-based collaborative perception dataset: CoPerception-UAVs, co-simulated by AirSim [34] and Carla [30], including 131.9K aerial images and 1.94M 3D boxes. The perception range is 200m×350m. For the camera-only 3D object detection task on CoPerception-UAVs, our detector follows DVDET [8]. The input aerial image size is $(800, 450)$. The aerial-view input feature map is transformed to BEV with the resolution of $0.25$m/pixel, and the size is $(192, 352)$; see more details in Appendix.

**DAIR-V2X.** DAIR-V2X [11] is the only public **real-world** collaborative perception dataset. Each sample contains two agents: a vehicle and an infrastructure, with 3D annotations. The perception range is 201.6m×80m. Originally DAIR-V2X does not label objects outside the camera's view, we relabel all objects to cover 360-degree detection range. We complement several intermediate fusion-based baselines on DAIR-V2X to comprehensively validate our method on real data. For LiDAR-based 3D object detection task, our detector follows PointPillar [35]. We represent the field of view into a BEV map with size $(200, 504, 64)$ and the resolution is $0.4$m/pixel in length and width.

## 5.2 Quantitative evaluation

**Benchmark comparison.** Fig. 3 compares the proposed `Where2comm` with the previous methods in terms of the trade-off between detection performance (AP@IoU=0.50) and communication bandwidth; also see exact values in Table 3 of Appendix. We consider single-agent detection without collaboration $(\widehat{\mathcal{O}}_i^{(0)})$, When2com [12], V2VNet [1], DiscoNet [2], V2X-ViT [26] and late fusion, where agents directly exchange the detected 3D boxes. The red curve comes from a single `Where2comm` model evaluated at varying bandwidths. We see that the proposed `Where2comm`: i) achieves a far-more superior perception-communication trade-off across all the communication bandwidth choices and various collaborative perception tasks, including camera-only 3D object detection from aerial view and car front view, and LiDAR-based 3D object detection; ii) achieves significant improvements over previous state-of-the-arts on both real-world (DAIR-V2X) and simulation scenarios, improves the SOTA performance by 7.7% on DAIR-V2X, 6.62% on CoPerception-UAVs, 25.81% on OPV2V, 1.9% on V2X-Sim; iii) achieves the same detection performance of previous state-of-the-arts with extremely less communication volume: 5128 times less on CoPerception-UAVs, more than 100K times less on OPV2V, 55 times less on V2X-Sim, 105 times less on DAIR-V2X.

**Multi-round evaluation.** Fig. 4 presents the performances of `Where2comm` at communication rounds ranging from 1 to 3. Each curve comes from a single `Where2comm` model with a certain communication round evaluated at varying bandwidths. Results show that 1 communication round is good, more rounds are even better. Multi-round communication steadily improves the performance-bandwidth trade-off across all three datasets, reflecting its effectiveness and robustness. This encourages the agents to actively collaborate without worrying the performance degradation. This also validates that `Where2comm` can well work at various communication bandwidths and rounds.

**Robustness to localization noise.** We follow the localization noise setting in V2VNet and V2X-ViT (Gaussian noise with a mean of 0m and a standard deviation of 0m-0.6m) and conduct experiments on all the three datasets to validate the robustness against realistic localization noise. *Where2comm* is more robust to the localization noise than previous SOTAs. Fig. 5 shows the detection performances as a function of localization noise level in CoPerception-UAVs, OPV2V and V2X-Sim datasets, respectively We see: i) overall the collaborative perception performance degrades with the increasing

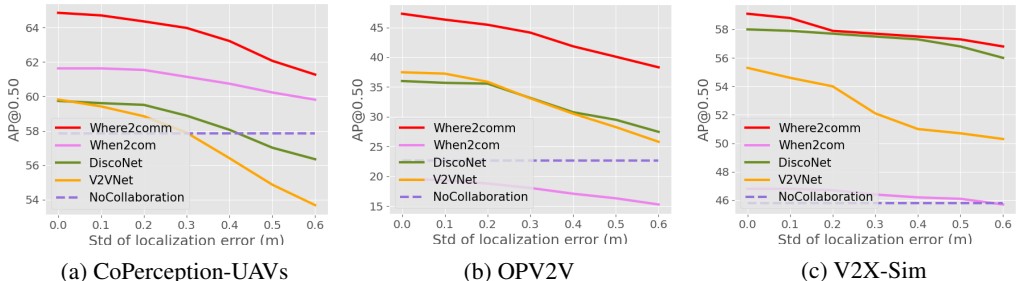

(a) CoPerception-UAVs       (b) OPV2V       (c) V2X-Sim

Figure 5: Robustness to localization error. Gaussian noise with zero mean and varying std is introduced. *Where2comm* consistently outperforms previous SOTAs and No Collaboration.

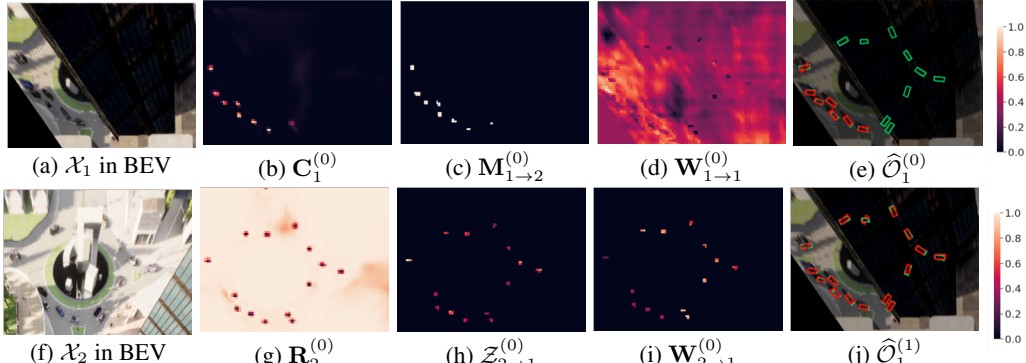

(a) $\mathcal{X}_1$ in BEV   (b) $\mathbf{C}_1^{(0)}$   (c) $\mathbf{M}_{1\to2}^{(0)}$   (d) $\mathbf{W}_{1\to1}^{(0)}$   (e) $\widehat{\mathcal{O}}_1^{(0)}$

(f) $\mathcal{X}_2$ in BEV   (g) $\mathbf{R}_2^{(0)}$   (h) $\mathcal{Z}_{2\to1}^{(0)}$   (i) $\mathbf{W}_{2\to1}^{(0)}$   (j) $\widehat{\mathcal{O}}_1^{(1)}$

Figure 6: Visualization of collaboration between Drone 1 and Drone 2 on CoPerception-UAVs dataset, including spatial confidence map ($\mathbf{C}_1^{(0)}$), selection matrix ($\mathbf{M}_{1\to2}^{(0)}$), message ($\{\mathbf{R}_2^{(0)}, \mathcal{Z}_{2\to1}^{(0)}\}$) in the communication module, attention weight in the fusion module ($\mathbf{W}_{1\to1}^{(0)}, \mathbf{W}_{2\to1}^{(0)}$), and Drone 1's detection results before ($\widehat{\mathcal{O}}_1^{(0)}$) and after ($\widehat{\mathcal{O}}_1^{(1)}$) collaboration. Green and red boxes denote ground-truth and detection, respectively. The objects occluded by a tall building can be detected through transmitting spatially sparse, yet perceptually critical message.

localization noise, while *where2comm* outperforms previous SOTAs (When2com, V2VNet,DiscoNet) under all the localization noise. ii) *where2comm* keeps being superior to *No Collaboration* while V2VNet fails when noise is over 0.4m and DiscoNet fails when noise is over 0.5m on CoPerception-UAVs. The reasons are: i) the powerful transformer architecture in fusion module attentively select the most suitable collaborative feature; ii) the spatial confidence map helps filter out noisy features, these two designs work together to mitigate noise localization distortion effects.

### 5.3 Qualitative evaluation

**Visualization of spatial confidence map.** Fig. 6 illustrates how `Where2comm` is empowered by the proposed spatial confidence map. In the scene, Drone 1's view is occluded by a tall building. With Drone 2's help, Drone 1 is able to detect through occlusion. Fig. 6 (a-d) shows Drone 1's observation, spatial confidence map (1), binary selection matrix (2), and ego attention weight (3). Fig. 6 (f-h) shows Drone 2's observation and message sent to Drone 1, including the request map (opposite of confidence map) and the sparse feature map, achieving efficient communication. Fig. 6 (i) shows the attention weight for Drone 1 to fuse Drone 2's messages, which is sparse, yet highlights the objects' positions. Fig. 6 (e) and (j) compares the detection results before and after the collaboration with Drone 2. We see that the proposed spatial confidence map contributes to spatially sparse, yet perceptually critical message, which effectively helps Drone 1 detect occluded objects.

**Visualization of detection results.** Fig. 7 shows that compared to *No Collaboration*, *When2com* and *DiscoNet*, `Where2comm` is able to achieves more complete and accurate detection results. The reason is that *When2com* employs a scalar to denote the agent-to-agent attention, which cannot distinguish which spatial area is more informative; *DiscoNet* employs a MLP-based fusion weight learning, which cannot well capture the complex collaboration attention; while `Where2comm` can zoom in to critical spatial areas in a cell-level resolution and leverage the spatial confidence map and sensing distances as priors to achieve more comprehensive fusion.

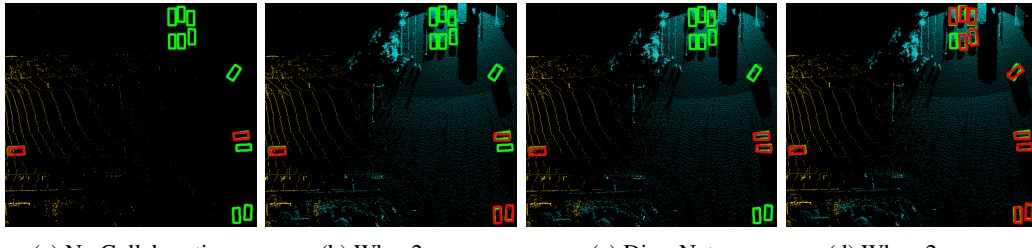

|           |            |            |              |
| :-------: | :--------: | :--------: | :----------: |
| (a) No Collaboration | (b) When2com | (c) DiscoNet | (d) Where2comm |

Figure 7: `Where2comm` qualitatively outperforms When2com and DiscoNet in DAIR-V2X dataset. Green and red boxes denote ground-truth and detection, respectively. Yellow and blue denote the point clouds collected from vehicle and infrastructure, respectively.

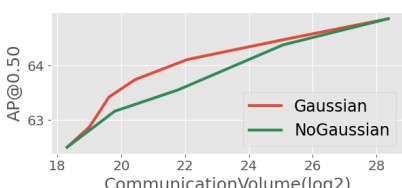

Figure 8: Selection matrix ablation study. Applying Gaussian filter improves performance.

Table 2: Fusion component ablation study. Multi-head attention (MHA), sensor positional encoding (SPE) and spatial confidence map (SCM) all improves the performances. Results are reported in AP@0.50/AP@0.70.

| MHA | SPE | SCM | OPV2V | CoPerception-UAVs | V2X-Sim |
| :-: | :-: | :-: | :---: | :---------------: | :-----: |
|     |     |     | 34.96/13.92 | 63.48/44.23 | 51.2/45.7 |
| ✓   |     |     | 38.75/13.28 | 63.99/44.46 | 57.3/50.8 |
| ✓   | ✓   |     | 39.82/16.43 | 64.34/46.86 | 59.1/52.0 |
| ✓   | ✓   | ✓   | **47.30/19.30** | **64.83/47.62** | **59.1/52.2** |

## 5.4 Ablation studies

**Effect of Gaussian filter in perceptually critical area selection.** Fig. 8 compares two versions of the selection matrix (2) with and without Gaussian filter. We see that applying Gaussian filter improves the overall performance. The reason is that: i) Gaussian filter could help filter out the outliers in the input map, selecting more robust critical regions; ii) it considers the context, benefiting the independent feature selection at each certain location by providing more information.

**Effect of components in spatial confidence-aware message fusion.** Tab. 2 assesses the effectiveness of the proposed fusion with two priors. We see that: i) per-location multi-head attention (MHA) outperforms the vanilla attention by 10.84% on OPV2V on AP@0.50, because MHA leverages information from multiple heads, better capturing cross-agent attention; and ii) As two informative priors, both sensing position encoding (SPE) and spatial confidence map (SCM) can consistently improve the performance. Especially, the version with all three designs improves the detection performance by 22.06% on OPV2V on AP@0.50.

## 6 Conclusion and limitation

We propose `Where2comm`, a novel communication-efficient collaborative perception framework. The core idea is to exploit a spatial confidence map at each agent to promote pragmatic compression, assisting agents to decide what to communicate with whom, and whose information to aggregate. Each agent offers spatially sparse, yet perceptually critical features to support other agents; meanwhile, requests complementary information from others in multi-round communication. Comprehensive experiments covering multi-type agents and multi-modality inputs show that `Where2comm` achieves far superior trade-off between perception performance and communication bandwidth.

**Limitation and future work.** The current work focuses on perceptually critical spatial areas. In future, we plan to expand a similar idea to the temporal dimension and determine critical time stamps. More cost will be reduced by exploring when to communicate. We also expect that more methods on pragmatic compression and emergent communication could be applied to collaborative perception.

**Acknowledgment.** This research is partially supported by the National Key R&D Program of China under Grant 2021ZD0112801, National Natural Science Foundation of China under Grant 62171276, the Science and Technology Commission of Shanghai Municipal under Grant 21511100900, CCF-DiDi GAIA Research Collaboration Plan 202112 and CALT Grant 2021-01.

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
