# Where2comm: Communication-Efficient Collaborative Perception via Spatial Confidence Maps

**Yue Hu**          **Shaoheng Fang**          **Zixing Lei**
Cooperative Medianet Innovation Center, Shanghai Jiao Tong University
`{18671129361, shfang, chezacarss}@sjtu.edu.cn`

**Yiqi Zhong**
University of Southern California
`yiqizhon@usc.edu`

**Siheng Chen**[*]
Shanghai Jiao Tong University, Shanghai AI Laboratory
`sihengc@sjtu.edu.cn`

## 1 Appendix

### 1.1 Highlights of our contribution

To sum up, our contributions are:

• We propose a novel fine-grained spatial-aware communication strategy, where each agent can decide where to communicate and pack messages only related to the most perceptually critical spatial areas. This strategy not only enables more precise support for other agents, but also more targeted request from other agents in multi-round communication.

• We propose `Where2comm`, a novel collaborative perception framework based on the spatial-aware communication strategy. With the guidance of the proposed spatial confidence map, `Where2comm` leverages novel message packing and communication graph learning to achieve lower communication bandwidth, and adopts confidence-aware multi-head attention to reach better perception performance.

• We conduct extensive experiments to validate `Where2comm` achieves state-of-the-art performance-bandwidth trade-off on multiple challenging real/simulated datasets across views and modalities.

### 1.2 Detailed information about the system pipeline

Alg. 1 presents the pipeline of our multi-round spatial confidence-aware collaborative perception system.

### 1.3 Detailed information about the optimization problem of collaborative perception

The constrained optimization in Sec.3 is the mathematical formation of collaborative perception. It is hard to obtain the global optimum due to hard constrains and non-differentialability of binary variables. Therefore, the proposed Where2comm essentially introduces an auxiliary variable and decomposes the original problem into two sub-optimization problems, each one of which is easy to solve.

To understand the details, let us consider a setting of fixed communication bandwidth and communication round, $K = 1, B = [B_1]$. Then, the optimization is

$$\max_{\theta,\mathcal{P}} \sum_{i=1}^{N} g\left(\Phi_\theta\left(\mathcal{X}_i, \{\mathcal{P}_{i\to j}\}_{j=1}^{N}\right), \mathcal{Y}_i\right), \text{ s.t. } \sum_{i,j=1}^{N} |\mathcal{P}_{i\to j}| \le B_1.$$

---

[*]Corresponding author

36th Conference on Neural Information Processing Systems (NeurIPS 2022).

**Algorithm 1** Multi-round spatial confidence-aware collaborative perception system

1: Define $N$ as the number of agents , $K$ as communication round
2: # Initialization
3: **for** $i = 1, 2, \ldots, N$, **do**
4:     $\mathcal{F}_i^{(0)} = \Phi_{\text{enc}}(\mathcal{X}_i) \in \mathbb{R}^{H \times W \times D}$                 ▷ Extract intermediate feature
5: **end for**
6: **for** $k = 0, 1, \ldots, K - 1$, **do**
7:     **for** $i = 1, 2, \ldots, N$, **do** # Each agent is computing individually
8:         $\mathbf{C}_i^{(k)} = \Phi_{\text{generator}}(\mathcal{F}_i^{(k)}) \in \mathbb{R}^{H \times W}$         ▷ Generate spatial confidence map
9:         **for** $j = 1, 2, \ldots, N$, **do**
10:             # Message packing
11:             $\mathbf{R}_i^{(k)} = 1 - \mathbf{C}_i^{(k)} \in \mathbb{R}^{H \times W}$           ▷ Pack request map
12:             **if** $k = 0$ **then**
13:                 $\mathbf{M}_{i \to j}^{(k)} = \Phi_{\text{select}}(\mathbf{C}_i^{(k)}) \in \{0, 1\}^{H \times W}$     ▷ Select critical areas
14:             **else**
15:                 $\mathbf{M}_{i \to j}^{(k)} = \Phi_{\text{select}}(\mathbf{C}_i^{(k)} \odot \mathbf{R}_j^{(k-1)}) \in \{0, 1\}^{H \times W}$   ▷ Select requested areas
16:             **end if**
17:             $\mathcal{Z}_{i \to j}^{(k)} = \mathbf{M}_{i \to j}^{(k)} \odot \mathcal{F}_i^{(k)} \in \mathbb{R}^{H \times W \times D}$       ▷ Pack spatially sparse features
18:             # Communication graph learning
19:             **if** $k = 0$ **then**
20:                 $\mathbf{A}_{i \to j}^{(k)} = 1$                ▷ Broadcast critical features and request
21:             **else**
22:                 $\mathbf{A}_{i \to j}^{(k)} = \max_{h,w} \left( \mathbf{M}_{i \to j}^{(k)} \right)_{h,w} \in \{0, 1\}$   ▷ Communicate only when necessary
23:             **end if**
24:         **end for**
25:         # Communication
26:         Send $\mathcal{P}_{i \to j} = \left( \mathcal{Z}_{i \to j}^{(k)}, \mathbf{R}_i^{(k)} \right)$ to other agents
27:         Receive $\{\mathcal{P}_{j \to i} = \left( \mathcal{Z}_{j \to i}^{(k)}, \mathbf{R}_j^{(k)} \right), j \neq i\}$ from other agents
28:         # Message fusion
29:         $\mathcal{F}_i^{(k+1)} = f_{\text{fuse}} \left( \mathcal{F}_i^{(k)}, \{(\mathcal{Z}_{j \to i}^{(k)}, \mathbf{R}_j^{(k)}), j = 1, 2, ..., N\} \right) \in \mathbb{R}^{H \times W \times D}$
30:     **end for**
31:     Store $\mathcal{F}_i^{(k+1)}$ and $\{\mathbf{R}_j^{(k)}, j \neq i\}$ for the next round
32: **end for**
33: $\mathcal{O}_i^{(K)} = \Phi_{\text{dec}}(\mathcal{F}_i^{(K)})$                   ▷ Output the final detections

Since there is only one round, we do not consider the request map, then, the message sent from the $i$th agent to the $j$th agent is $\mathcal{P}_{i \to j} = \mathcal{Z}_{i \to j} = \mathbf{M}_{i \to j} \odot \mathcal{F}_i$, whose spatial sparsity is determined by the binary selection mask $\mathbf{M}_{i \to j}$. Note that $\mathbf{M}_{i \to j}$ determines where to communicate, and is the key of the proposed Where2comm. Then, the original optimization is equivalent to

$$\max_{\theta, \mathbf{M}} \sum_{i=1}^N g \left( \Phi_\theta \left( \mathcal{X}_i, \{\mathbf{M}_{i \to j}\}_{j=1}^N \right), \mathcal{Y}_i \right), \ \ \text{s.t.} \sum_{i=1}^N \sum_{j=1, j \neq i}^N |\mathbf{M}_{i \to j}| \leq b_1, \mathbf{M}_{i \to j} \in \{0, 1\}^{H \times W},$$
(1)

where $\mathcal{F}_i$ can attribute to the network $\Phi_\theta(\cdot)$ and input data $\mathcal{X}_i$ and $b_1 = B_1/D$ with $D$ the channel number of $\mathcal{F}_i$. Due to the binary constrains, it is hard to optimize (1) directly. Instead, we decompose (1) into two sub-optimization problems and optimize the binary selection matrix $\mathbf{M}_{i \to j}$ and the network parameters $\theta$ once at a time: i) obtain a feasible binary selection matrix $\mathbf{M}_{i \to j}$ by optimizing a proxy constrained problem; ii) given the feasible binary selection matrix $\mathbf{M}_{i \to j}$, optimize the perception network parameter $\theta$. The constraint is satisfied in i) and the perception goal is achieved in ii). Specifically, two sub-optimization problems are

● **Obtain a feasible binary selection matrix $\mathbf{M}_{i \to j}$.** This essentially optimizes where to allocate the communication bandwidth. Intuitively, the spatial confidence reflects the perceptually critical level,

so that those spatial regions with higher spatial confidence will provide more critical information to help the partners and should have a higher priority be selected.

Following this spirit, we consider a proxy constrained problem as follows,

$$\max_{\mathbf{M}} \sum_{i=1}^{N} \sum_{j=1, j \neq i}^{N} \mathbf{M}_{i \to j} \odot \mathbf{C}_i, \ \text{ s.t. } \sum_{i=1}^{N} \sum_{j=1, j \neq i}^{N} |\mathbf{M}_{i \to j}| \leq b_1, \mathbf{M}_{i \to j} \in \{0, 1\}^{H \times W},$$

where $\mathbf{C}_i$ is the spatial confidence map. Note that i) even this optimization problem has hard constraints and non-differentialability of binary variables, it has an analytical solution that naturally satisfies all the constraints in (1); and ii) even we cannot solve the original objective, this proxy objective still carries the similar idea to promote better, yet more compact perception. This solution is obtained by selecting those spatial regions whose corresponding elements in $\mathbf{M}$ rank top-$b_1$. The detailed steps of **selection function** are: i) arrange the elements in the input matrix in descending order; ii) given the communication budget constrain, decide the total number ($b_1$) of communication regions; iii) set the spatial regions of $\mathbf{M}$, where elements rank in top-$b_1$ as the 1 and 0 verses.

● **Given the feasible binary selection matrix, optimize the network parameter $\theta$.** This essentially optimizes the perception performance. The sub-problem is

$$\max_{\theta} \sum_{i=1}^{N} g \left( \Phi_\theta \left( \mathcal{X}_i, \{\mathbf{M}_{i \to j}\}_{j=1}^N \right), \mathcal{Y}_i \right).$$

This can be solved by standard **supervised learning**. For example, the perception evaluation metric $g(\cdot)$ can be evaluated by the detection loss calculated between detections and the ground-truth and the detection loss is optimized with an Adam optimizer. We thus get the optimized perception network parameter $\theta$. Note that this sub-problem does not involve any constraints and is thus easy to optimize.

### 1.4 Detailed information about the module design

**Observation encoder.** Here we elaborate on the warping functions for the monocular camera, where the depth is unknown and estimated. Instead of directly projecting 2D features to flat ground space, we first lift them to 3D voxel space and then collapse them to the BEV. This design considers all the possible depths/altitudes, introducing flexibility in the projection, and mitigating the distortion effect caused by information loss in imaging. The detailed steps are: **1)** Categorical Depth Distribution Network (CaDDN [1]), which is a recent and effective method to warp image feature to BEV feature, is applied to estimate the depth distribution for each image feature point. **2)** Each feature point is wrapped from the 2D image space to the 3D physic space according to the known camera parameters. **3)** The 3D voxel features are flattened to BEV features. Briefly, the warping function is unfolded as follows: for each image feature point locates at $(u, v)$, given the estimated categorical depth $d_i$, and the known camera projection matrix $\mathbf{P} \in \mathbb{R}^{3 \times 4}$, 3D physical space coordinates $[x, y, z]^T$ is calculated conditioned on the image feature coordinates $[u, v, d_i]^T$ based on the projection function: $[u, v, d_i]^T = \mathbf{P} \cdot [x, y, z, 1]^T$.

**Spatial confidence-aware message packing.** Fig. 1 presents the detail about the spatial confidence-aware message packing module. For the message from agent $i$ to agent $j$ at $k$th communication round, the module takes the spatial confidence map $\mathbf{C}_i^{(k)}$ of agent $i$ and the request map $\mathbf{R}_j^{(k-1)}$ of agent $j$ as input, and outputs the message $\mathcal{P}_{i \to j}^{(k)}$ including the masked feature map $\mathcal{Z}_{i \to j}^{(k)}$ and the request map of agent $i$.

**Spatial confidence-aware communication graph construction.** Fig. 2 presents the comparisons on the communication graph with previous works. *Fully connected* versus *agent-level partially connected* versus ours *spatial-decouple partially connected* communication. *Fully connected* communication results in a large amount of bandwidth usage, growing on the order of $O(N^2)$, where N is the number of agents in a network. *Agent-level partially connected* communication prune irrelevant connections between agents while may erroneously sever the information connection. *Spatial-decouple partially connected* communication could further flexibly prune irrelevant connections per-location and can substantially reduce the overall network complexity.

**Spatial confidence-aware message fusion.** Fig. 3 presents the detail about the spatial confidence-aware message fusion module. Given the received messages $\{\mathcal{P}_{j \to i}^{(k)}, j \in \mathcal{N}_i\}$, each agent $i$ attentively

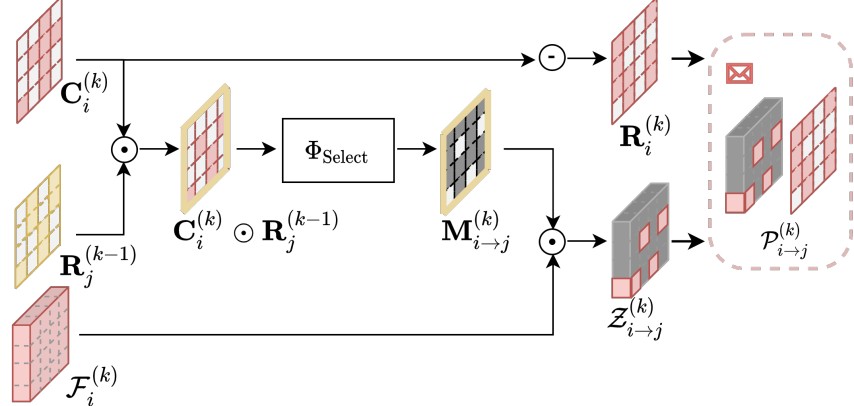

Figure 1: Spatial confidence-aware message packing module. $\odot$ denotes point-wise multiplication, $\ominus$ denotes point-wise minus by a matrix with the same shape as the input and filled with 1. Best viewed in color. Grey denotes the location being filled with zeros for the binary selection matrix $\mathbf{M}_{i \to j}^{(k)}$ and the feature map $\mathcal{Z}_{i \to j}^{(k)}$.

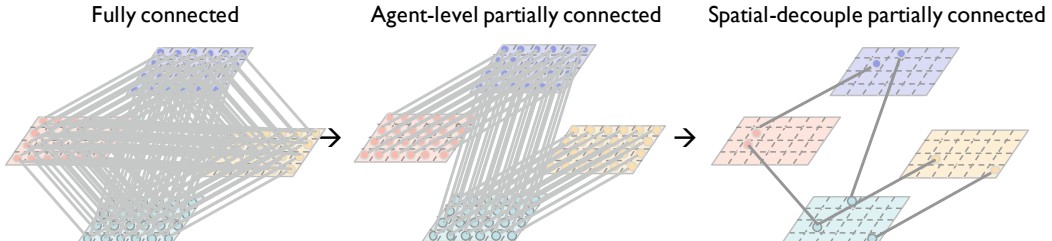

Figure 2: Spatial confidence-aware communication graph construction module. We spatially decouple the full feature map, and could flexibly involve the informative spatial areas in the communication. This *Spatial-decouple partially connected* communication could further flexibly prune irrelevant connections per-location and is more bandwidth-efficient.

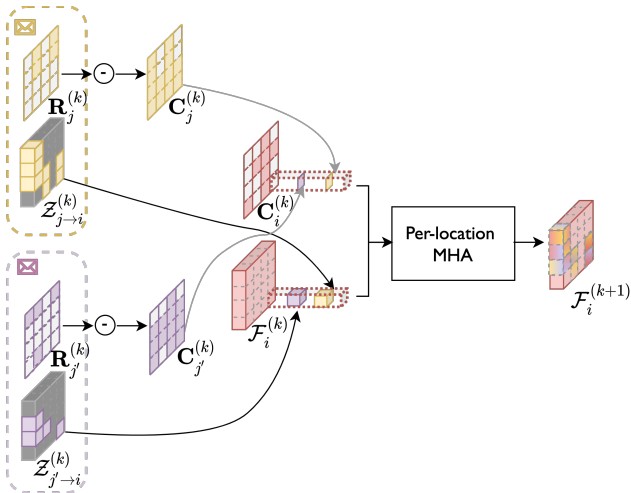

Figure 3: Spatial confidence-aware message fusion module. Each agent attentively augments the features with the received messages at each location. And the per-location multi-head attention are separately operated at each location in parallel, it takes the features and the corresponding confidence scores as input, and outputs the augmented features.

Table 1: Overall performance on CoPerception-UAVs, OPV2V, V2X-Sim and DAIR-V2X. Comm denotes the communication volume calculated with Equation (2).

| Dataset | CoPerception-UAVs | | OPV2V | | V2X-Sim1.0 | | DAIR-V2X | |
|---|---|---|---|---|---|---|---|---|
| Method/Metric | Comm | AP@0.50/0.70 | Comm | AP@0.50/0.70 | Comm | AP@0.50 | Comm | AP@0.50/0.70 |
| No Collaboration | 0.00 | 57.67/29.52 | 0.00 | 22.65/9.09 | 0.00 | 45.80 | 0.00 | 50.03/43.57 |
| Late Fusion | 15.77 | 53.12/37.88 | 11.87 | 8.24/3.84 | 8.83 | 46.70 | 11.45 | 53.12/37.88 |
| When2com | 28.37 | 61.63/33.55 | 22.28 | 19.69/8.29 | 20.00 | 46.70 | 22.62 | 51.12/36.17 |
| V2VNet | 29.95 | 59.82/33.14 | 23.87 | 37.47/14.67 | 21.58 | 55.30 | 24.21 | 56.01/42.25 |
| V2X-ViT | 28.37 | 59.12/41.57 | 22.28 | 39.82/16.43 | 20.00 | 57.30 | 22.62 | 54.26/43.35 |
| DiscoNet | 28.37 | 59.74/29.71 | 22.28 | 36.00/12.50 | 20.00 | 58.00 | 22.62 | 54.29/44.88 |
| | 11.76 | 60.19/34.94 | 5.67 | 40.11/15.36 | 6.70 | 47.60 | 11.40 | 50.98/39.11 |
| | 14.27 | 60.23/34.93 | 15.49 | 42.15/16.09 | 8.29 | 49.10 | 15.58 | 51.01/39.10 |
| | 15.73 | 61.30/35.29 | 16.13 | 43.37/16.84 | 9.52 | 50.60 | 17.03 | 53.53/40.70 |
| | 17.96 | 63.04/36.10 | 17.04 | 44.07/17.15 | 10.41 | 51.80 | 17.53 | 55.84/42.44 |
| **Where2comm** | 19.04 | 63.94/37.16 | 17.86 | 44.68/17.77 | 11.10 | 54.20 | 18.19 | 58.46/44.46 |
| | 21.62 | 65.10/38.98 | 18.43 | 45.23/18.02 | 12.27 | 56.60 | 20.56 | 63.54/48.78 |
| | 23.33 | 65.32/39.25 | 18.92 | 46.04/18.23 | 12.80 | 57.00 | 21.78 | 63.76/48.94 |
| | 25.31 | 65.46/39.27 | 18.92 | 46.04/18.23 | 13.98 | 58.90 | 22.35 | 63.71/48.89 |
| | 28.48 | 65.71/39.38 | 22.71 | 47.14/19.07 | 20.00 | 59.10 | 22.62 | 63.71/48.93 |

augments the features with the received messages at each location. And the request map $\mathbf{R}_j^{(k)}$ in the received message is firstly decoded to the confidence map $\mathbf{C}_j^{(k)}$ via a point-wise minus. Then the per-location multi-head attention are separately operated at each location in parallel, it takes the features and the corresponding confidence scores as input, and outputs the augmented features.

**Sensor positional encoding.** Sensor positional encoding is conditioned on the physical distance between the known sensor coordinates and each BEV gird's coordinate in the 3D physic space. It is introduced to provide spatial prior, as the smaller the sensing distance is, the clear the observation would be. Mathematically, similar to the position encoding in [2], our sensor positional encoding is given by $SPE_{(dis,2p)} = sin(dis/10000^{2p/D}), SPE_{(dis,2p+1)} = cos(dis/10000^{2p/D})$ where $dis$ is the physical distance, $p$ is the dimension, $D$ is the total channel dimension of the BEV feature map, $sin$ and $cos$ denote the sine and cosine functions.

### 1.5 Detailed information about experimental settings

**Implementation details.** For camera-only 3D object detection task on OPV2V, we implement the detector following CADDN [1]. The model is trained 100 epoch with initial learning rate of 1e-3, and decay by 0.1 at epoch 80. For LiDAR-based 3D object detection task, our detector follows MotionNet [3]. We train 120 epoch with learning rate 1e-3. For the camera-only 3D object detection task on CoPerception-UAVs, our detector follows the CenterNet [4] with DLA-34 [5] backbone. The model is trained 140 epoch with learning rate 5e-4.

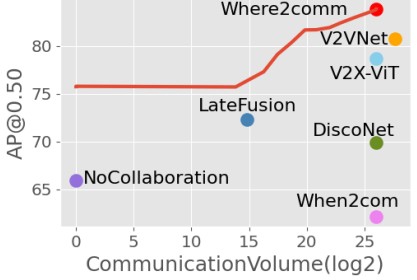

Figure 4: Where2comm achieves consistently superior performance-bandwidth trade-off on V2X-Sim2.0 [6].

Table 2: Overall performance on V2X-Sim2.0 [6]. Comm denotes the communication volume calculated with Equation( 2). Metric AP@(0.50/0.70) is used.

| Method/Metric | Comm | AP@0.50/0.70 | Method | Comm | AP@0.50/0.70 |
|---|---|---|---|---|---|
| No Collaboration | 0.00 | 65.93/51.79 | | 13.84 | 75.72/65.13 |
| Late Fusion | 14.84 | 72.33/62.12 | | 17.47 | 79.14/67.05 |
| When2com | 26.04 | 62.15/49.42 | **Where 2comm** | 19.84 | 81.69/70.79 |
| V2VNet | 27.62 | 80.80/71.22 | | 21.98 | 81.94/72.10 |
| V2X-ViT | 26.04 | 78.73/63.17 | | 24.13 | 82.99/73.05 |
| DiscoNet | 26.04 | 69.73/55.12 | | 25.93 | 83.77/74.09 |

**Inference strategy in multi-round setting.** For the single-round communication, all the communication budget are used in this broadcast communication round. For the two-round communication, a small bandwidth (about 20%) is allocated to activate the collaboration; for the next round, the remained relatively large (about 80%) bandwidth is allocated to transmit the targeted information to meet agents' request. For more than two rounds communication setting, we strategically allocate

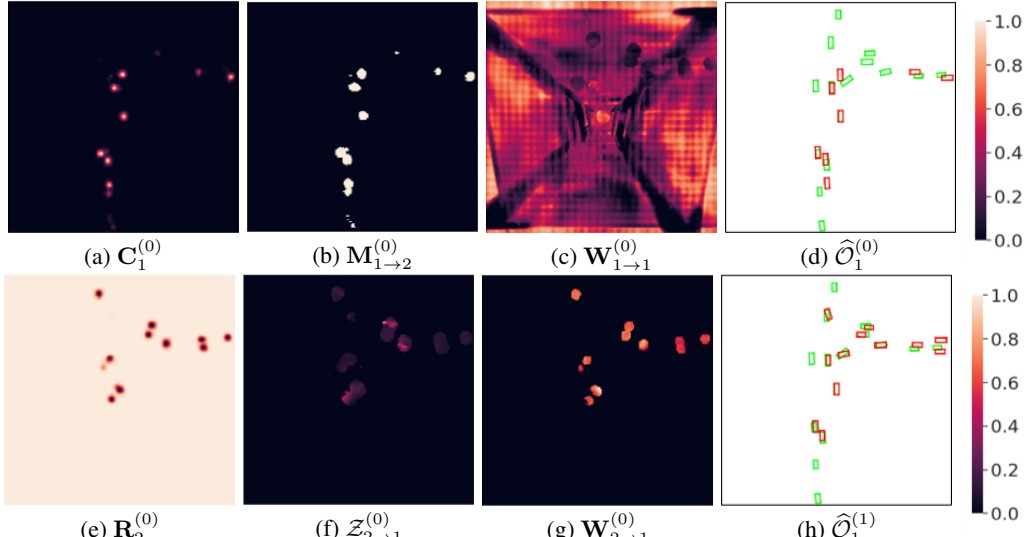

Figure 5: Visualization of collaboration between Vehicle 1 and Vehicle 2 on OPV2V dataset, including spatial confidence map ($\mathbf{C}_1^{(0)}$), selection matrix ($\mathbf{M}_{1\to2}^{(0)}$), message ($\{\mathbf{R}_2^{(0)}, \mathcal{Z}_{2\to1}^{(0)}\}$) in the communication module, attention weight in the fusion module ($\mathbf{W}_{1\to1}^{(0)}, \mathbf{W}_{2\to1}^{(0)}$), and Vehicle 1's detection results before ($\widehat{\mathcal{O}}_1^{(0)}$) and after ($\widehat{\mathcal{O}}_1^{(1)}$) collaboration. Green and red boxes denote ground-truth and detection, respectively. The objects occluded can be detected through transmitting spatially sparse, yet perceptually critical message.

communication budget across multiple communication rounds. For the initial broadcast round, a small bandwidth (about 20%) is allocated to activate the collaboration; for the next round, a relatively large (about 60%) bandwidth is allocated to transmit the targeted information to meet agents' request; then, the bandwidth is gradually reduced, accounting for the communication degradation with the increasing rounds.

**Communication volume.** Our communication volume is the same as DiscoNet [7], the only difference is that our log base is 2, while it is 10, so our number is about 3.32 times theirs. The base 2 is chosen to align with the metric bit/byte, this is, communication volume counts the message size by byte in log scale with base 2. Mathematically for the selected sparse feature map $\mathcal{Z}_{i\to j}^{(k)} = \mathbf{M}_{i\to j}^{(k)} \odot \mathcal{F}_i^{(k)} \in \mathbb{R}^{H \times W \times D}$, the communication volume is

$$\log_2\left(|\mathbf{M}_{i\to j}^{(k)}| \times D \times 32/8\right), \tag{2}$$

where $|\cdot|$ denotes the L0 norm counting the non-zero elements in the binary selection matrix, this is, the total spatial girds need to be transmitted, and for each feature point $D$ denotes the channel dimension, 32 is multiplied as float32 data type is used to represent each number, 8 is divided as the metric byte is used.

## 1.6 Benchmarks

We conduct extensive experiments on all the available collaborative perception benchmarks. Tab. 1 presents the overall performance on the four datasets, CoPerception-UAVs, OPV2V [8], V2X-Sim1.0 [7] and DAIR-V2X [9]. And we further benchmark the updated V2X-Sim2.0 [6] in Fig. 4 and Tab. 2. For this LiDAR-based 3D object detection task, our detector follows PointPillar [10]. We see that where2comm consistently achieves significant improvements over previous methods on all the benchmarks.

## 1.7 Visualization

**Visualization of collaboration in OPV2V and V2X-Sim.** Fig. 5 and Fig. 6 illustrates how Where2comm is empowered by the proposed spatial confidence map on OPV2V and V2X-Sim dataset. In the scene, with Vehicle 2's help, Vehicle 1 is able to detect the missed objects in

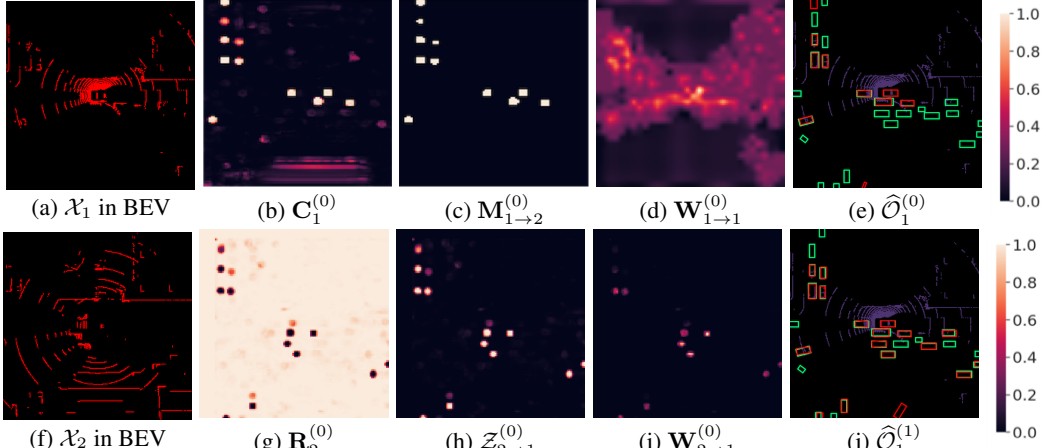

(a) $\mathcal{X}_1$ in BEV    (b) $\mathbf{C}_1^{(0)}$    (c) $\mathbf{M}_{1\to 2}^{(0)}$    (d) $\mathbf{W}_{1\to 1}^{(0)}$    (e) $\widehat{\mathcal{O}}_1^{(0)}$

(f) $\mathcal{X}_2$ in BEV    (g) $\mathbf{R}_2^{(0)}$    (h) $\mathcal{Z}_{2\to 1}^{(0)}$    (i) $\mathbf{W}_{2\to 1}^{(0)}$    (j) $\widehat{\mathcal{O}}_1^{(1)}$

Figure 6: Visualization of collaboration between Vehicle 1 and Vehicle 2 on V2X-Sim dataset, including spatial confidence map ($\mathbf{C}_1^{(0)}$), selection matrix ($\mathbf{M}_{1\to 2}^{(0)}$), message ($\{\mathbf{R}_2^{(0)}, \mathcal{Z}_{2\to 1}^{(0)}\}$) in the communication module, attention weight in the fusion module ($\mathbf{W}_{1\to 1}^{(0)}, \mathbf{W}_{2\to 1}^{(0)}$), and Drone 1's detection results before ($\widehat{\mathcal{O}}_1^{(0)}$) and after ($\widehat{\mathcal{O}}_1^{(1)}$) collaboration. Green and red boxes denote ground-truth and detection, respectively. The objects occluded by a tall building can be detected through transmitting spatially sparse, yet perceptually critical message.

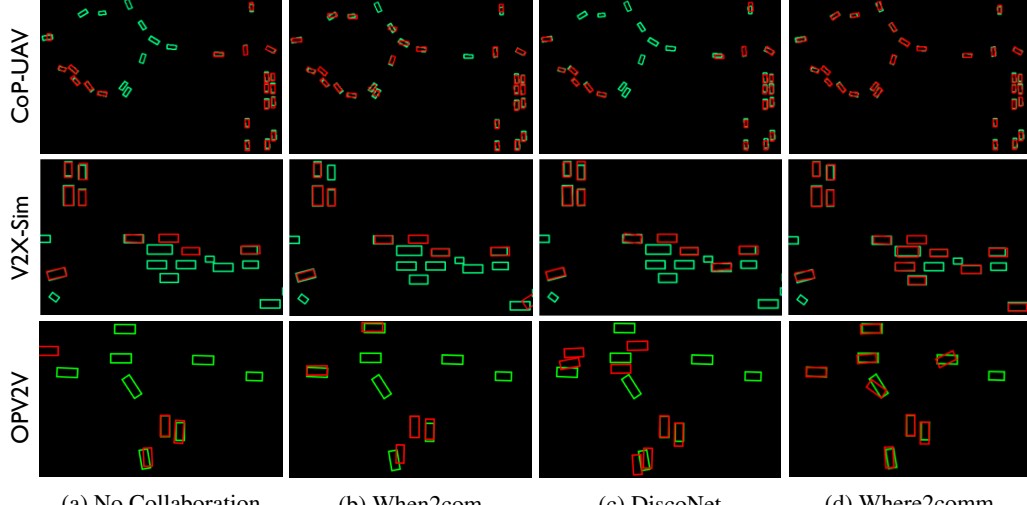

(a) No Collaboration    (b) When2com    (c) DiscoNet    (d) Where2comm

Figure 7: `Where2comm` qualitatively outperforms the state-of-the-art methods in CoPerception-UAVs, V2X-Sim and OPV2V datasets. Green and red boxes denote ground-truth and detection, respectively.

the single view. Fig. 5 (a-d) shows Vehicle 1's spatial confidence map, binary selection matrix, ego attention weight, and the detection results by its own observation. Fig. 5 (e-f) shows Vehicle 2's message sent to Drone 1, including the request map (opposite of confidence map) and the sparse feature map, achieving efficient communication. Fig. 5 (g) shows the attention weight for Vehicle 1 to fuse Vehicle 2's messages, which is sparse, yet highlights the objects' positions. Fig. 5 (d) and (h) compares the detection results before and after the collaboration with Vehicle 2. We see that the proposed spatial confidence map contributes to spatially sparse, yet perceptually critical message, which effectively helps Vehicle 1 detect occluded objects.

**Visualization of detection results.** Fig. 7 shows that `Where2comm` qualitatively outperforms the state-of-the-art methods in CoPerception-UAVs, V2X-Sim and OPV2V datasets.

### 1.8    Ablation on bandwidth allocation

Fig. 8 shows the bandwidth allocation ablation study in multi-round communication setting. We see that allocating more bandwidth in the second and subsequent communication rounds achieves a better

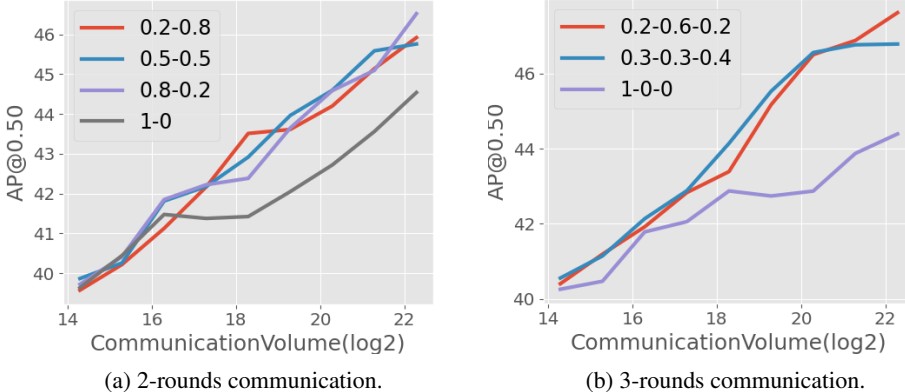

|  (a) 2-rounds communication. |  (b) 3-rounds communication. |

Figure 8: Bandwidth allocation ablation study in multi-round communication. (a-b) shows the perception performance and communication bandwidth trade-offs for 2- and 3-round communication using different bandwidth allocation strategies on the OPV2V dataset. The legend shows the bandwidth ratio from the initial communication round to the entire communication round. Allocating more bandwidth in the second and subsequent communication rounds achieves a better performance-bandwidth trade-off than allocating all bandwidth in the initial communication round.

performance-bandwidth trade-off than allocating all bandwidth in the initial communication round, and the gain is stable for different bandwidth allocation strategies. The reason is that multi-round communication employs a request map in the second and subsequent communication rounds to denote the spatial area where each agent needs more information, which enables more targeted and efficient communication.

## 1.9   Discussion on the realistic limitations

There are many challenges in a collaborative perception system. In this work, we focus on the biggest challenge in current collaborative perception systems; that is, the trade-off between communication bandwidth and perception performance. This challenge has been actively addressed in previous works [11–13, 7]. Because collaborative perception is enabled and also severely limited by the communication capacity, which is critically reflected in the highly dynamic and limited bandwidth in real-world communication systems. *Where2comm* flexibly adapts to various communication bandwidths, achieving superior performance-bandwidth trade-off.

Here we further discuss other realistic limitations, assess the robustness of our system and future improvements to be done.
• For other realistic communication issues such as **latency**, *where2comm* communicates strategically when necessary, rather than all the time or everywhere, to reduce the possibility of encountering communication problems. In addition, a prediction module could be integrated to estimate the missed or delayed frames according to the historically received frames. And by focusing on the informative spatial regions, *where2comm* can reduce the estimation difficulty.
• For the **time synchronisation** issue, by using the powerful transformer architecture-based fusion module, *where2comm* can attentively augment the features with the received asynchronous features from other agents. In addition, *where2comm* can introduce positional encoding conditioned on delay time and easily extend to global multi-head attention to further reduce the effects of time synchronization.
• For the **noisy localization** issue, *where2comm* exchanges the intermediate features among agents, which has a relatively low spatial resolution, thus is relatively robust to noisy pose. In addition, *where2comm* can easily extend to a deformable transformer architecture like [14] to further alleviate the feature distortion caused by the noisy localization.
• For the **attack** issue, by focusing on specific spatial regions and attentively fusing the received features from other agents, *where2comm* is relatively less likely to be attacked.
• For the **data availability**, *where2comm* works on both RGB and point cloud modalities, and is sensor friendly, so it can be deployed on cheap camera sensors and lidar sensors.

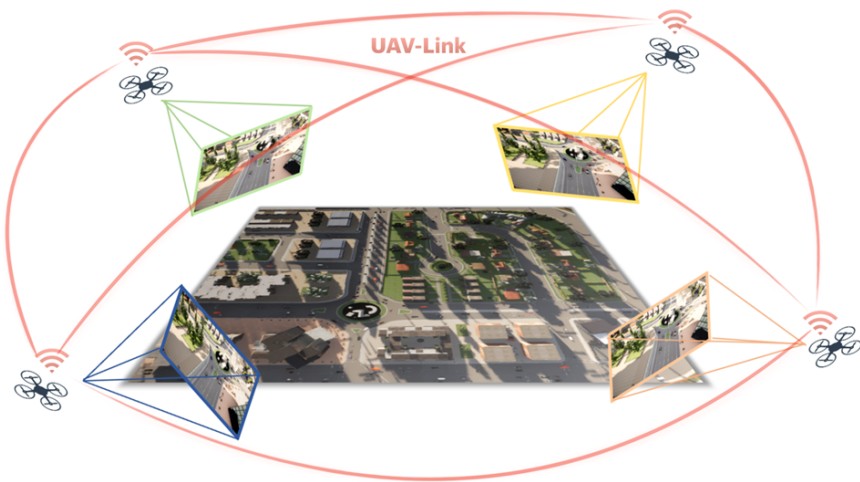

Figure 9: As an important component of the UAV swarm, collaborative perception could fundamentally resolve various reception-field restrictions in the traditional single-agent perception.

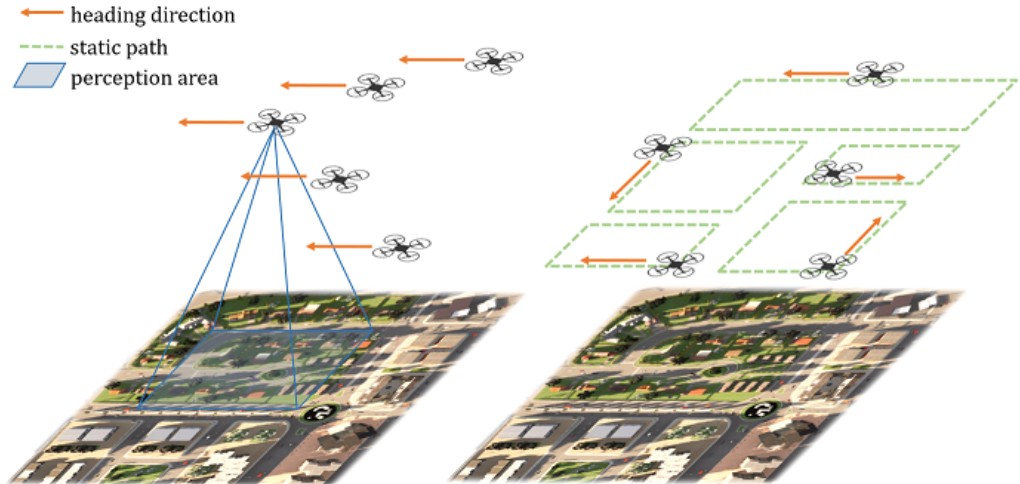

Figure 10: Two types of UAV swarm formation. The left shows the discipline formation mode, where the swarm keeps a static array and the right shows the dynamic formation mode, where each UAV navigates independently in the scene.

## 1.10  CoPerception-UAVs dataset details

CoPerception-UAVs dataset collects data from drones, see Fig. 9. As the rapid development of an unmanned aerial vehicle (UAV) significantly enhances human's ability to perceive the world from an aerial perspective. UAV-based systems have been widely used in numerous applications, including search and rescue, security and surveillance, photography, geographical mapping, as well as traffic monitoring. Through collaboration, UAV swarm can further distribute multiple tasks and achieve higher flexibility, stronger robustness, and a larger perception range, leading to significant advantages in harsh and complex environments. Unfortunately, collaborative perception mainly focuses on the vehicles and ignores the UAV literature. To **provide more diverse views and challenging benchmark for the collaborative perception community**, here we present the first comprehensive large-scale collaborative perception dataset for UAV swarm so far.

Since building a dataset in the real world is too expensive and laborious, in this initial version, we consider a virtual dataset based on the co-simulation of AirSim [15] and Carla [16], where AirSim simulates the UAV swarms and Carla simulates the complex background scenes and dynamic foreground objects. In the simulation, we consider that the UAV swarm is flying over diverse simulated scenes at various altitudes. Each UAV has a sensing device to collect RGB images, a computation device to perceive the environment with a perception model, and a communication

device to transmit perception information among UAVs. In this setting, the UAV swarm is able to achieve 2D/3D object detection, pixel-wise or bird's-eye-view (BEV) semantic segmentation in a collaborative manner. Our dataset consists of 131.9k synchronous images collected from 5 coordinated UAVs flying at 3 altitudes over 3 simulated towns with 2 types of swarm formation. To enable the model training and testing, each image is fully annotated with the pixel-wise semantic segmentation labels, 2D bounding boxes of vehicles, as well as 3D bounding boxes on the ground and the semantic mask from BEV view. This benchmark can enable the evaluation of collaborative perception methods on the important perception tasks: 2D/3D/BEV object detection and semantic segmentation. The dataset details as unfolded as follows.

**Data collection.** Our proposed dataset is collected by the co-simulation of CARLA [16] and AirSim[15] (both under MIT license). We use CARLA to generate complex simulation scenes and traffic flow; and use AirSim to simulate UAV swarm flying in the scene. The flight route of UAVs is controlled by AirSim and sample data are collected randomly at about 4-second intervals.

**Map creation.** The simulation scenes, including the road layout, static objects, and traffic flow, are created based on CARLA [16] simulation. We take three open-source maps (*town4* to *town6*) provided by CARLA as the basic road layouts, which are the three largest maps in scale. To increase the complexity and diversity of the scenes and make the perception tasks more challenging, we customize the original maps, adding and replacing various buildings, vegetation, roadblocks, barriers, and other static objects with various assets provided by CARLA.

**Traffic flow creation.** Moving vehicles in the scene are managed through CARLA. Hundreds of vehicles are spawned in each scene by script *spawn_npc.py* provided by CARLA. The initial location and motion trajectory of each vehicle is determined by the map's road layout.

**Sensor setup.** Each UAV is equipped with 5 RGB cameras in 5 directions and 5 semantic cameras collecting semantic ground truth for RGB cameras. The cameras include a bird's eye view camera and four cameras facing forward, backward, right, and left with a pitch degree of $-45°$. Each camera has an FoV of $90°$ and the resolution is $800 \times 450$. On each UAV, all the cameras are fixed and their internal relative position and rotation degree are invariable. The translation (x, y, z) and rotation (w, x, y, z in quaternion) of each camera in both global and ego coordinates are recorded during data collection. With such a sensor setting, a UAV at the height of $40m$ can mostly cover an area of $200m \times 200m$.

**Formation flying.** The UAV swarm moves and executes tasks in the three-dimensional space, where the situation could be much more complex than those of vehicles or roadside units. In our proposed dataset, we take into consideration two main factors that may affect the perception and collaboration patterns of UAV swarms: flight formation and altitude. Each UAV swarm consists of 5 UAVs. We arrange two types of formation modes for a UAV swarm: discipline mode, where all 5 UAVs keeps a consistent and relatively static array, and dynamic mode, where each UAV navigates independently in the scene; see Fig. 10. The former simulates the situation where the swarm of UAVs is executing a same specific task such as exploring an unknown area, search and rescue; while the latter simulates the monitoring and patrolling tasks in the city.

Fully-annotated data are provided in our proposed dataset, including synchronous images with pixel-wise semantic labels, 2D & 3D bounding boxes of vehicles, and BEV semantic map; see 11.

**Camera data.** We collect synchronous images from all cameras on 5 UAVs, which is 25 images in a sample. Camera intrinsics and extrinsics in global coordinate are provided to support coordinate transformation across various UAVs. In total, 123.8K images are collected for the discipline swarm mode and 8.1K for the dynamic swarm mode.

**Bounding boxes.** During data collection, 3D bounding boxes of vehicles are recorded at the same moment with images, including location (x, y, z), rotation (w, x, y, z in quaternion) in the global coordinate and their length, width and height. The location (x, y, z) is the center of the bounding box. Then we provide 2D bounding boxes by projecting the 3D bounding boxes to the image perspective plane of each camera, resulting in 1.94M 3D bounding boxes and 3.6M 2D bounding boxes in total.

**Data usage.** In total, CoPerception-UAVs has 131.9K aerial images and 1.94M 3D boxes. We randomly split the samples into train/validation/test, resulting 91,175/19,500/20,250 images, and 1,316,536/303,888/319,576 3D bounding boxes. The dataset is organized in a similar way with

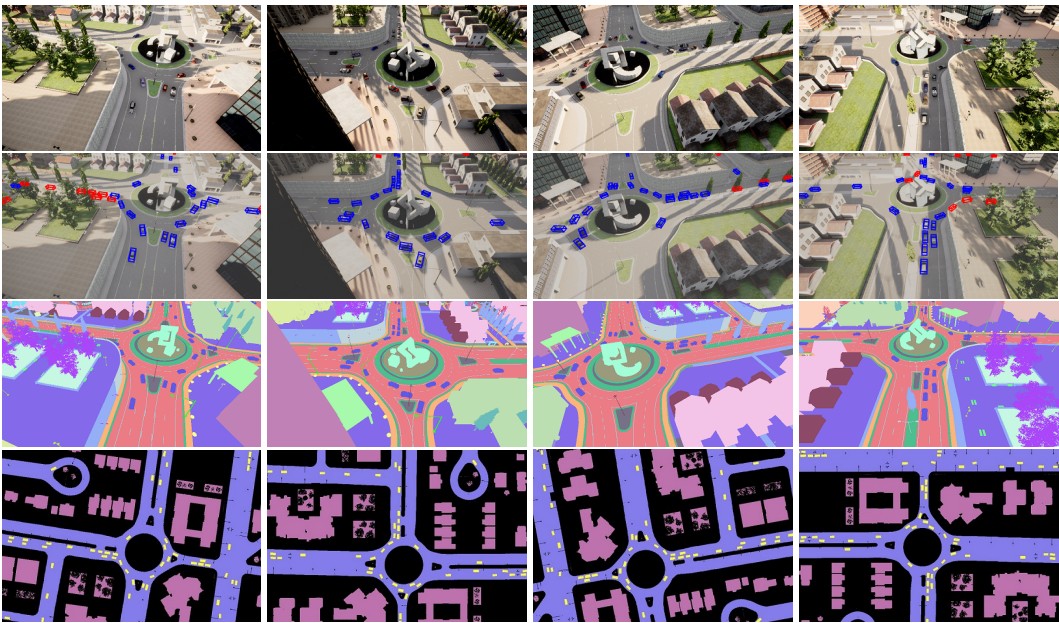

Figure 11: Data and annotations of one sample. From top to bottom: RGB image, image with 3D bounding boxes, image with semantic labels, and BEV map with semantic labels. From left to right are from different cameras equipped on four UAVs.

the widely-used autonomous driving dataset, nuScenes [17]; so it can be used directly with the well-established nuScenes-devkit.