# OpenReview forum: "Where2comm: Communication-Efficient Collaborative Perception via Spatial Confidence Maps"
_NeurIPS.cc/2022/Conference — NeurIPS 2022 Accept_

### Official Review · Reviewer_Jgi5 · 2022-07-09

**Rating:** 6
**Confidence:** 4
**Soundness:** 2 fair
**Presentation:** 2 fair
**Contribution:** 3 good

**Summary:**

This paper studies the task of multi-agent collaborative perception with the purpose of achieving better trade-off between performance and bandwidth consumption. To reduce the communication cost across the agents, the authors propose a new approach to generate and transmit the spatial confidence map instead of the traditional whole feature map.  The framework proposed in this paper consists of five modules, including the observation encoder, spatial confidence generator, spatial confidence-aware communication module, communication graph construction, spatial confidence-aware message fusion and detection decoder. The authors have conducted experiments on three datasets to verify the effectiveness of their new approach. Compared with five baselines, the new approach has shown higher detection accuracy and smaller communication cost.

**Questions:**

1. Where are the results of the five baselines from?  Did the authors run the public source codes of the baselines on the three datasets?  Could the authors provide some details in the paper?

2. In the original paper, DiscoNet has reported detection accuracy of 60.3% and 58.5% in two different versions on dataset V2X-Sim, without and with 16 times compression, respectively. However, the accuracy of DiscoNet is around 58.0% in this paper. Why the results in two papers have the difference?

3. Where is the dataset CoPerception-UAVs from? It seems it's not a public dataset. If so, more details on the dataset would be helpful. For example, collecting data from drones is much trickier than from self-driving cars since drones can have any arbitrary poses. So more details are required. Also, how is training/val/test conducted? The appendix doesn’t contain dataset details even though the main text claims so.

4. How are the constrained loss function and binary variables optimized?

5. Can you elaborate more on "sensor positional encoding"? Why should it represent the physical distance between each agent’s sensor and its observation? There is neither justifications nor references provided.

6. Given the complexity of the proposed method, it would be great if the authors could provide the code (the same as the baseline methods). So people check for details that aren't provided from the paper.


**Limitations:**

Yes

**Strengths And Weaknesses:**

Strengths:
1. It is a good idea to transmit part of the feature map across the agents to reduce the communication volume, and potentially improve detection accuracy.
2. The authors have conducted extensive experiments on three different datasets to validate the new approach. The experiments cover two types of agents (cars and drones), and two types of sensors (LiDAR and camera). They also present many experimental results to support the conclusion, including quantitative comparison and visualization.

Weaknesses:
1. It’s a very complex model with 5 modules. Although the authors present the high-level ideas of each module, a lot of details on the architectures and optimization techniques are missing. For example, how the constrained optimization presented in Sec. 3 is optimized? How are binary variables in M (Eq. 2) and A optimized? Are there any optimization issues due to hard constrains and non-differentialability of binary variables?

2. The authors compare their proposed method against five baselines. I have checked the original papers of the baselines, and found that only DiscoNet reports the results on V2X-Sim, which is one of three datasets in this paper. Since most results of five baselines reported in this work are not directly from the original papers, a solid comparison between the proposed approach and the baselines is required. Apparently, Fig. 3(c) shows that the performance gap between DiscoNet and Where2Com on V2X-Sim is much smaller than the other two datasets, while V2X-Sim is the only dataset that DiscoNet reported its performance.

3. The performance of communication volume varies much on three different datasets, 5128 times less on CoPerception-UAVs, 10,000 times less on OPV2V, and 55 times less on V2X-Sim. It seems that the authors do not introduce the details of the metric for communication volume. I am not sure whether the baselines used the same metric before. The comparison also has the potential fairness issue.

4. Overall, given the presentation of the paper and a large amount of details are missing, I don’t think the algorithm can be easily reproduced by readers.

---

> ### Author Response · Authors · 2022-08-01
> **Response to Reviewer Jgi5's comments on [Question 3]**
>
> ### [Q3:] More details about CoPerception-UAVs? How is training/val/test conducted?
>
> Sorry for missing the details! Here are some details about CoPerception-UAVs; **see more in the updated supplementary material**.
>
> **1)** We will release CoPerception-UAVs dataset, which collects data from drones. Unmanned aerial vehicle (UAV)-based systems have significantly enhanced human’s ability to perceive the world and are widely used in numerous applications, including search, rescue, and traffic monitoring. Through collaboration, UAV swarms can achieve higher flexibility, stronger robustness, and a larger perception range, leading to significant advantages in complex environments. Unfortunately, collaborative perception mainly focuses on vehicles and ignores UAVs. To **provide more diverse views and challenging benchmarks for the collaborative perception community**, we present the first comprehensive large-scale collaborative perception dataset for the UAV swarm so far.
>
> **2)** As building a real record dataset is too expensive and laborious, we consider a virtual dataset based on the co-simulation of AirSim and Carla. The UAV swarm is flying over diverse simulated scenes at various altitudes. This datset consists of 131.9k synchronous images collected from 5 coordinated UAVs flying at 3 altitudes over 3 simulated towns with 2 types of swarm formation. And it can support the evaluation of collaborative perception methods on the important perception tasks: 2D/3D/BEV object detection and semantic segmentation. The details unfolded as follows.
>
> **Data collection** This dataset is collected by the co-simulation of CARLA and AirSim. We use CARLA to generate complex simulation scenes and traffic flow, and use AirSim to simulate UAV swarm flying in the scene. The flight route of UAVs is controlled by AirSim and sample data are collected randomly at about 4-second intervals.
>
> **Map creation.** The simulation scenes, including the road layout, static objects, and traffic flow, are created based on CARLA simulation. We take the three largest open-source maps (*town4* to *town6*) provided by CARLA as the basic road layouts. To increase the complexity and diversity of the scenes and make the perception tasks more challenging, we customize the original maps, adding and replacing various buildings, vegetation, roadblocks, barriers, and other static objects with various assets provided by CARLA.
>
> **Traffic flow creation.** Moving vehicles in the scene are managed through CARLA. Hundreds of vehicles are spawned in each scene by script *spawn\_npc.py* provided by CARLA. The initial location and motion trajectory of each vehicle is determined by the map's road layout.
>
> **Sensor setup.** Each UAV is equipped with 5 RGB cameras in 5 directions and 5 semantic cameras collecting semantic ground truth for RGB cameras. The cameras include a bird's eye view camera and four cameras facing forward, backward, right, and left with a pitch degree of $-45^\circ$. Each camera has an FoV of $90^\circ$ and the resolution is $800\times450$. On each UAV, all the cameras are fixed and their internal relative position and rotation degree are invariable. The translation and rotation of each camera in both global and ego coordinates are recorded during data collection.
>
> **Formation flying.** The UAV swarm moves and executes tasks in the 3D space, where the situation could be much more complex than those of vehicles or roadside units. We arrange two types of formation modes for a UAV swarm: discipline mode, where all UAVs keep a consistent and relatively static array, and dynamic mode, where each UAV navigates independently in the scene. The former simulates the situation where the swarm of UAVs is executing the same specific task such as exploring an unknown area, search and rescue; while the latter simulates the monitoring and patrolling tasks in the city.
>
> We provide fully-annotated data, including synchronous images with pixel-wise semantic labels, 2D & 3D bounding boxes of vehicles, and BEV semantic map.
>
> **Camera data.** We collect synchronous images from all cameras on 5 UAVs, which is 25 images in a sample. Camera intrinsics and extrinsic in global coordinate are provided to support coordinate transformation across various UAVs. In total, 123.8K images are collected for the discipline swarm mode and 8.1K for the dynamic swarm mode.
>
> **Bounding boxes.** 3D bounding boxes of vehicles are recorded at the same moment with images, including location, rotation in the global coordinate, and their length, width, and height. Then we provide 2D bounding boxes by projecting the 3D bounding boxes to the image perspective plane of each camera, resulting in 1.94M 3D bounding boxes and 3.6M 2D bounding boxes in total.
>
> **3)** In total, CoPerception-UAVs has 131.9K aerial images and 1.94M 3D boxes. We randomly split the samples into train/validation/test, resulting 91,175/19,500/20,250 images, and 1,316,536/303,888/319,576 3D bounding boxes.

---

> > ### Comment · Reviewer_Jgi5 · 2022-08-08
> > **Thank you for the responses**
> >
> > Thank you for your time and effort in this response. I have read the updated paper and all the rebuttal. All my major concerns have been addressed. I have updated my rating accordingly.
> >
> > This is a much improved version as many missing details are now incorporated. Given the complexity of the algorithm, more details need to be checked and verified from the code. Thanks for the excellent work.

---

> > > ### Author Response · Authors · 2022-08-08
> > > **Thank you for the feedback**
> > >
> > > Thanks very much for reading our responses and providing your feedback! As we promised, if the submission gets accepted, we will release the codes, model checkpoints, and the dataset as soon as possible.

---

> ### Author Response · Authors · 2022-08-01
> **Response to Reviewer Jgi5's comments on [Weakness 3-4 & Question 5-6]**
>
> ### [Weakness 3:] The details of the metric for communication volume.
>
> Sorry for the confusion!
> 1) Communication volume is mentioned on Line 256 in the paper and is also relevant to Figure 2 in the supplementary material. It counts the message size by byte in log scale with base $2$.
> 2) Mathematically for the selected sparse feature map $\mathcal{Z}\_{i\rightarrow j}^{(k)}=\mathbf{M}\_{i\rightarrow j}^{(k)}\odot \mathcal{F}\_i^{(k)} \in \mathbb{R}^{H\times W \times D}$, the communication volume is $\text{log}\_2\left(|\mathbf{M}\_{i\rightarrow j}^{(k)}| \times D \times 32 / 8\right)$, where $|\cdot|$ denotes the L0 norm counting the non-zero elements in the binary selection matrix, this is, the total spatial girds need to be transmitted, and for each feature point $D$ denotes the channel dimension, $32$ is multiplied as float32 data type is used to represent each number, $8$ is divided as the metric byte is used.
> 3) Our communication volume is the same as DiscoNet, the only difference is that our log base is $2$, while it is $10$, so our number is about $3.32$ times theirs.
>
> We have updated the supplementary material to reflect this.
>
> ### [Weakness 4 & Q6:] If details/codes could be provided?
>
> We appreciate the reviewer's interest in our details and codes! We fully understand the reviewer’s concern, because V2VNet (ECCV 2020) does not show the dataset and the code even now, which makes some hard time for others to reproduce their work. But **we will definitely release all the source codes, and model checkpoints including our methods and previous baselines on all three datasets, as well as the CoPerception-UAVs dataset**. So, the readers will easily reproduce and further improve the system.
>
> ### [Q5:]  Why "sensor positional encoding" represents the physical distance between each agent’s sensor and its observation?
>
> **1)** By definition, sensor positional encoding is conditioned on the physical distance between the known sensor coordinates and each BEV gird's coordinate in the 3D physic space. It is introduced to provide spatial prior, as the smaller the sensing distance is, the clear the observation would be.
>
> **2)**  Mathematically, similar to the position encoding in [1], our sensor positional encoding is given by
> $$
> SPE_{(dis, 2p)} = sin(dis/10000^{2p/D}),\\\\
> SPE_{(dis, 2p+1)} = cos(dis/10000^{2p/D})
> $$
> where $dis$ is the physical distance, $p$ is the dimension, $D$ is the total channel dimension of the BEV feature map, $sin$ and $cos$ denote the sine and cosine functions.
>
> We have updated the supplementary material to reflect this.
>
> [1] Vaswani, Ashish et al. “Attention is All you Need.” ArXiv abs/1706.03762 (2017): n. pag.

---

> ### Author Response · Authors · 2022-08-01
> **Response to Reviewer Jgi5's comments on [Weakness 2 & Question 2]**
>
> ### [Weakness 2 \& Q2:]   Where are the results of the five baselines from? Why the results on V2X-Sim is different from DiscoNet?  Why performance gap is smaller on V2X-Sim?
>
> We really appreciate the reviewer’s efforts in careful reading! In short, our results, including DiscoNet on V2X-Sim, are the latest and convincing. We elaborate on three aspects:
>
> **1)** We did put a lot of effort to present solid comparisons between the proposed method and many baseline methods on diverse datasets and settings. As shown in the following table, previous works only reported their results on a single dataset with a single view and modality, while where2comm is validated on three datasets, including all the released 3D collaborative perception benchmarks (V2X-Sim and OPV2V) and a self-organized dataset (CoPerception-UAVs), covering diverse views and modalities. In terms of the dataset, we implement the dataloader based on the official code of each dataset. In terms of the baseline methods, When2com and DiscoNet have released their codes, so we directly use the official codes;  V2VNet does not release the code, we reproduce the implementation based on the code provided by DiscoNet; and we further implement NoCollaboration, LateFusion and V2V. For the two modality inputs, we accordingly implement the pre-processing, as images need to be warpped to BEV additionally.
>
> | Method                     | Venue     | Dataset           | Released               | Real/Simulated            | View                     | Perception | Modality                |
> |----------------------------|-----------|-------------------|------------------------|---------------------------|--------------------------|------------|-------------------------|
> | V2VNet                     | ECCV2020    | V2V-Sim           | X                | Simulated                 | Vehicle                  | 3D Det.    | PointCloud              |
> | Who2com                    | ICRA2020    | Airsim-Map        | &#10004;                 | Simulated                 | Drone                    | 2D Seg.    | Image                   |
> | When2com                   | CVPR2020    | Airsim-Map        | &#10004;                | Simulated                 | Drone                    | 2D Seg.    | Image                   |
> | AAOMAC                     | ICCV2021  | V2V-Sim           | X                | Simulated                 | Vehicle                  | 3D Det.    | PointCloud              |
> | DiscoNet                   | Neurips2021 | V2X-Sim           | &#10004;                | Simulated                 | Vehicle                  | 3D Det.    | PointCloud              |
> | OPV2V                      | ICRA2022    | OPV2V             | &#10004;                | Simulated                 | Vehicle                  | 3D Det.    | PointCloud              |
> | V2X-ViT                    | ECCV2022    | OPV2V             | &#10004;                | Simulated                 | Vehicle                  | 3D Det.    | PointCloud              |
> | SyncNet                    | ECCV2022    | V2X-Sim           | &#10004;                | Simulated                 | Vehicle                  | 3D Det.    | PointCloud              |
> | Where2comm | Submit to Neurips2022 | V2X-Sim, OPV2V, CoPerception-UAVs  | &#10004; | Simulated | Vehicle & Drone}     | 3D Det. | Image & PointCloud} |
>
>
> **2)**  **The difference between performance reported in DiscoNet and ours on V2X-Sim is caused by dataset bug fixes and updates by the V2X-Sim authors**. V2X-Sim is a new dataset and has been constantly fixing some bugs and finetuned. The author releases V2X-Sim 1.0 (Oct. 2021). And we download and conduct experiments on the released dataset, and exchange data usage experiences and provide recommendations for dataset improvements. The author provides us with their latest updated dataset and best checkpoints. We use the latest version by the time of submission, and the performance of 58.0\% reported in the paper comes from the best checkpoint provided by the author. Recently, the author releases version 2.0 (June 2022). We will update the results in the revised version.
>
> **3)** As Tab.3 in DiscoNet shown, the improvement on V2X-Sim is in a relatively smaller range, and the closer to upper-bound, the harder it to improve. DiscoNet reports 2.8 gain over simple concatenation fusion, here *where2comm* achieves 1.0 gain over DiscoNet, achieving more close performance to the upper bound (early fusion, 64.2\%). Furthermore, the other two datasets with monocular input show more significant improvements. The reason is that: for monocular 3D detection, transforming features from 2D to 3D encounters distortion issues, which leads to more challenging feature aggregation and also makes a larger gap between lower-bound and upper-bound, and the strength of the transformer in fusion is more pronounced.

---

> ### Author Response · Authors · 2022-08-01
> **Response to Reviewer Jgi5's comments on [Weakness 1b & Question 4]**
>
> ### [Weakness 1b & Q4:] How optimization (Sec. 3), M (Eq. 2) and A optimized? Are there any optimization issues due to hard constraints and non-differentiability?
> Thanks for this insightful question! As the reviewer mentioned, it is hard to obtain the global optimum of the constrained optimization in Sec.3 due to hard constraints and non-differentiability of binary variables. Therefore, **from a mathematical perspective, the proposed Where2comm essentially introduces an auxiliary variable and decomposes the original problem into two sub-optimization problems, each one of which is easy to solve.**
>
> To understand the details, let us consider a setting of fixed communication bandwidth and round, $K=1,B=[B\_1]$. Then, the optimization is
> $$\underset{\theta,\mathcal{P}}{\max}\sum\_{i=1}^{N} g \left(\Phi\_{\theta} \left(\mathcal{X}\_i,\\{\mathcal{P}\_{i\rightarrow j} \\}\_{j=1}^N \right), \mathcal{Y}\_i  \right),{\rm s.t.} \sum\_{i,j=1}^{N}|\mathcal{P}\_{i\rightarrow j}| \leq B_1.$$
> Since there is only one round, we do not consider the request map, then, the message sent from the $i$th agent to the $j$th agent is $\mathcal{P}\_{i\rightarrow j}=\mathcal{Z}\_{i\rightarrow j}=\mathbf{M}\_{i\rightarrow j} \odot \mathcal{F}\_i,$ whose spatial sparsity is determined by the binary selection mask $\mathbf{M}\_{i\rightarrow j}$. Note that $\mathbf{M}\_{i\rightarrow j}$ determines where to communicate, and is the key of the proposed Where2comm. Then, the original optimization is equivalent to
> $$ \underset{\theta,\mathbf{M}}{\max}\sum\_{i=1}^{N} g \left(\Phi\_{\theta} \left(\mathcal{X}\_i,\\{\mathbf{M}\_{i\rightarrow j}\\}\_{j=1}^N \right), \mathcal{Y}\_i  \right),{\rm s.t.} \sum\_{i=1}^{N}\sum\_{j=1,j\neq i}^{N}|\mathbf{M}\_{i\rightarrow j}| \leq b\_1, \mathbf{M}\_{i\rightarrow j}\in\\{0,1\\}^{H\times W},\tag{1}$$
> where $\mathcal{F}\_i$ can attribute to the network $\Phi\_{\theta}(\cdot)$ and input data $\mathcal{X}\_i$ and $b\_1 = B\_1/D$ with $D$ the channel number of $\mathcal{F}\_i$. Due to the binary constraints, it is hard to optimize (1) directly. Instead, we decompose (1) into two sub-optimization problems and optimize the binary selection matrix $\mathbf{M}\_{i\rightarrow j}$ and the network parameters $\theta$ once at a time. The constraint is satisfied in Problem 1. and the perception goal is achieved in Problem 2. Specifically, two sub-optimization problems are:
>
> **1)** **Obtain a feasible binary selection matrix $\mathbf{M}\_{i\rightarrow j}$ by optimizing a proxy constrained problem.** This essentially optimizes where to allocate the communication bandwidth. Intuitively, the spatial confidence reflects the perceptually critical level, so those spatial regions with higher spatial confidence will provide more critical information to help the partners and should have a higher priority be selected.
>
> Following this spirit, we consider a proxy constrained problem as follows,
> $$ \underset{\mathbf{M}}{\max}\sum_{i=1}^{N} \sum\_{j=1,j\neq i}^{N}\mathbf{M}\_{i\rightarrow j}\odot\mathbf{C}\_i,{\rm s.t.}\sum\_{i=1}^{N}\sum\_{j=1,j\neq i}^{N}|\mathbf{M}\_{i\rightarrow j}| \leq b\_1, \mathbf{M}\_{i\rightarrow j}\in\\{0,1\\}^{H\times W},\tag{2}$$
> where $\mathbf{C}\_i$ is the spatial confidence map. Note that i) even though this optimization problem has hard constraints and non-differentiability of binary variables, it has an analytical solution that naturally satisfies all the constraints in (1); and ii) even we cannot solve the original objective, this proxy objective still carries the similar idea to promote better, yet more compact perception. This solution is obtained by selecting those spatial regions whose corresponding elements in $\mathbf{M}$ rank top-$b\_1$. As stated in Line 164, the detailed steps are:
> * Arrange the elements in the input matrix in descending order;
> * Given the communication budget constrain, decide the total number ($b\_1$) of communication regions;
> * Set the spatial regions of $\mathbf{M}$, where elements rank in top-$b\_1$ as the $1$ and $0$ verses.
>
> **2)** **Given the feasible binary selection matrix $\mathbf{M}\_{i\rightarrow j}$, optimize the perception network parameter $\theta$.** This essentially optimizes the perception performance. The sub-problem is
> $$\underset{\theta}{\max}\sum\_{i=1}^{N}g\left(\Phi\_{\theta}\left(\mathcal{X}\_i,\\{\mathbf{M}\_{i\rightarrow j}\\}_{j=1}^N\right),\mathcal{Y}\_i \right).$$
> This can be solved by standard supervised learning. For example, the perception evaluation metric $g(\cdot)$ can be evaluated by the detection loss calculated between detections and the ground-truth, and the detection loss is optimized with an Adam optimizer. We thus get the optimized perception network parameter $\theta$. Note that this sub-problem does not involve any constraints and is thus easy to optimize.
>
> **We have updated the supplementary material to reflect this.**

---

> ### Author Response · Authors · 2022-08-01
> **Response to Reviewer Jgi5's comments on [Weakness 1a & Question 1]**
>
> ### [Weakness 1a & Q1:] Complex model? A lot of details are missing.
>
> We fully understand and respect the reviewer's concern about complexity and details. Based on to all the reviewers's questions, **we have significantly revised the supplementary material to include as many details as we can.** The updated part is marked in blue. To further resolve the reviewer's concern, **we will definitely release all the source codes, model checkpoints, and CoPerception UAV dataset.** So the readers will know the details not only from the words but also from our code.
>
> Meanwhile, we want to address three facts: **i)** the task of collaborative perception itself is challenging and needs multiple modules; **ii)** the number of modules does not necessarily mean that a model is complex, and **iii)** we have tried as much as we can to provide more details in both the main paper and the supplementary materials. We elaborate those facts as follows.
>
> **i)** To build a collaborative perception system based on feature-level fusion, conceptually, at least four modules are required:  1) an observation encoder (we have to extract perceptual features); 2) communication module (we have to share information to other agents); 3) message fusion module (we have to fuse other agents; information); and 4) a decoder (we have to obtain the final output.) All the previous works, including Who2com (ICRA 2020), When2com (CVPR 2020), V2VNet (ECCV 2020) and DiscoNet (NeurIPS 2021), have these four modules. The technical contributions of previous works come from designing one or more modules in better ways. In this work, our motivation is very simple: exploiting the spatial heterogeneity of perceptual information to achieve better trade-off between perception performance and communication bandwidth. To implement this, we introduce one additional module, spatial confidence generator, which generates a spatial confidence map. Such a spatial confidence map is consumed by the communication module and the message fusion module to reduce the communication volume and improve the perception performance. Therefore, **multiple modules are not due to our design; it comes from the scale of this task.**
>
> **ii)** Even though the model has five modules, **each module is simple and the overall implementation is straightforward.**  In Figure 2, the overall pipeline and the data-flow between modules are clearly stated. We also have summarized all the implementation steps to Algorithm 1 in the supplementary material. The implementation of a spatial confidence generator is simply a common detection decoder, which produces the detection confidence map; the implementation of message packing is a selection function, which is simply a score-ranking operation, see Line 167 in the paper; the implementation of communication graph construction is simply one if-else statement; see Line 197 in the paper; the implementation of message fusion is a transformer-architecture; the implementation of detection decoder is conventional, which decodes the features into objects. And the overall loss function follows the detection loss and the optimized by an Adam optimizer. We have been always trying to present our method as clearly as possible through multiple forms of graphics, algorithms, and text. Therefore, we believe that too many modern architectures are way more complex than our design and the complexity is not an obvious weakness of the proposed method.
>
> **iii)** Like many machine learning papers, we admit that we are not able to show all the details due to the page limit. But we did address a lot of details in the original supplementary material. For example, Section 1.2 shows the detailed information about the system pipeline; Section 1.3 shows the detailed information about the module design, and Section 1.4 shows the experimental settings. **We have great enthusiasm and have been always trying our utmost to present a high-quality work.**

---

### Official Review · Reviewer_YUZ9 · 2022-07-10

**Rating:** 6
**Confidence:** 3
**Soundness:** 3 good
**Presentation:** 3 good
**Contribution:** 3 good

**Summary:**

This paper proposes a framework, where2comm, to enable multi-agent collaborative perception via a more improved information sharing through communications among agents. The core idea of where2comm is the use of a spatial confidence map, enabling each agent to decide which area needs to be shared and communicated about with other agents. Where2comm is composed of (1) spatial confidence generator (2) spatial confidence-aware communication module, (3) spatial confidence-aware message fusion module. Figure 2 well elaborates the overall structure of the system; the camera/lidar input goes through the encoder first, and the output of the encoder goes through the spatial confidence generator, which generates spatial confidence map that shows which areas in the map is perceptually critical level. The use of spatial confidence map is leveraged during the communication phase, which selects where to communicate about.

**Questions:**

Q1. The select function in the message packing module: to me, select function is not well described in the section. Selection function takes the element-wise  multiplication of confidence map of i-th agent C_i and request map of j-th agent R_j as inputs, and returns the output by selecting the largest elements of the given input. What does largest elements mean? Under which criteria the inputs of the C_i*R_j survive?

Q2. Communication graph is constructed from the adjacency matrix A_ij^k, which is the result of the maximum value of the binary selection matrix M_ij^k. It seems that A_ij^k is matrix is binary (0 or 1) matrix; what is the form of M_ij^k matrix? Are A and M both binary matrix? Can you visualize M matrix and A matrix?

Q3. Figure 5 (visualization) is very helpful. (b) is the confidence map, which indicates that the drone1’s view is more confidence about the non-dark area in the (a). How should I interpret (g)? It seems that this is the request map from the drone2 view. What does the red spots in the (g) request map represent?

**Limitations:**

The authors mentioned about the limitations, which is that this paper only focuses on perceptually critical spatial areas, and doesn't care about temporal dimension. I think this is reasonable limitations and suggestions for the future work.

**Strengths And Weaknesses:**

Strength of this paper comes from the originality of the framework, especially novel modules like message packing and communication graph construction. Message packing includes request map, which indicates where (which areas) needs to be further known, and is computed from negative of the confidence score. Also, there is spatially sparse map, which are computed from requests from other agents.

Communication graph construction identifies when and who to communicate by measuring overlap between information i-th agent has and information j-th agent needs.

I think the overall strength of this paper comes with sound problem formulation, the need for a fraemwork for multi-agent collaborative perception, and original/novel framework proposals. Weak points (or the points that need to be clarified) will be asked in the Questions section.

Another strength is that the where2comm achieves thousand times less communication volume and still outperforms other baselines, (Figure 3).

---

> ### Author Response · Authors · 2022-08-01
> **Response to Reviewer YUZ9's comments**
>
> ### [Q1] How the select function work?
>
> The function of the binary selection matrix is to select spatial regions for communication. The steps are:
> 1) arrange the elements in the input matrix in descending order;
> 2) given the communication budget constrain, decide the total number ($b_1$) of communication regions;
> 3) set the spatial regions of $\mathbf{M}$, where elements rank in top-$b_1$ as the $1$ and $0$ verses.
>
>
> ### [Q2] What is the form of $M_{ij}^{(k)}$ matrix? Are A and M both binary matrix? Can you visualize M and A?
>
> **i)** The selection matrix $M_{ij}^{(k)}\in\\{0,1\\}^{H\times W}$ is **binary matrix**, the same shape as the feature map. And $1$ represents that the spatial region should be transmitted, and vice versa. $A_{ij}^{(k)}$ is a **binary scalar** which is **the element of the binary communication graph's adjacency matrix** $A^{(k)}\in\\{0,1\\}^{N\times N}$, and $1$ represents that communication is built from agent $i$ to agent $j$, and vice versa.
>
> **ii)** The definition could be found in equation (2) and Line 197 in the paper. For the initial communication round ($k=0$), to motivate all the agents to participate in the collaboration, communications are built among all the agents, and the selection matrix selects the sparse features to be transmitted; for the following communication round ($k>1$), to promote the communication efficiency, communications are built among those agents who have features to be transmitted, which is decided by the selection matrix.
>
> **iii)** $M_{ij}^{(k)}\in\\{0,1\\}^{H\times W}$ matrix is visualized in Figure 5 (c), and as M matrix is not all-zero, this is, there is information to be transmitted; thus the communication from agent $i$ to agent $j$ should be built, this is, $A_{ij}^{(k)}$ is $1$.
>
> ### [Q3] How to interpret (g)? What do the red spots in the (g) request map represent?
>
> **1)** (b) is the confidence map, which indicates that drone1’s view is more confident about the **areas with objects** in the (a).
>
> **2)** (g) is the request map, negatively correlated with the confidence map. The darker the smaller. The dark red dots indicate lower request scores, this is, the drone2’s view has **less missing information in the areas with objects** in (f). The intuition is that: for the locations with low detection confidence scores, an agent is hard to tell if there are really no objects or it is just caused by the limited information (e.g. occlusion), thus, a high request score is set to ask for more complementary information from other agents, and vice versa.

---

### Official Review · Reviewer_VaH1 · 2022-07-11

**Rating:** 6
**Confidence:** 4
**Soundness:** 3 good
**Presentation:** 3 good
**Contribution:** 3 good

**Summary:**

Multi-agent collaborative perception could avoid accidents due to an agent's limited field of view via sensing information from other agents. However, the communication bandwidth is a bottleneck for multi-agent collaborative perception systems. The critical question is what to send and whom to send it. The authors propose a novel framework called Where2comm for 3D detection to address the challenges. A spatial confidence map is proposed to advance the performance of multi-agent collaborative perception. The paper benchmarks the framework on three datasets, OPV2V, V2X-Sim, and our original CoPerception-UAVs. The framework achieves state-of-the-art performance on these benchmarks. Moreover, the authors show that the proposed method can significantly reduce communication bandwidth while maintaining favorable performance compared with STOA.

**Questions:**

The paper presents a promising framework for Multi-agent collaborative perception for 3D object detection. The authors motivate the readers and propose a spatial confidence map that can be used to communicate critical information with other agents and reduce communication bandwidth. The experiments empirically demonstrate the effectiveness of the proposed framework. The reviewer recommends accepting the paper. The current rating is weak accept (6). The reviewer's main concern is the sensitivity of the framework to noisy localization, which might significantly degrade the contribution of a spatial confidence map in a real-world setting.

**Limitations:**

The authors did mention limitations of the work on the lack of considering the temporal aspect of critical timestamps.

**Strengths And Weaknesses:**

Strengths:
1. To reduce the number of traffic fatalities, multi-agent collaborative perception for 3D object detection is a promising direction, and it is a topic of interest to machine learning, computer vision, and intelligent transportation system researchers.
2. To the best knowledge of the reviewer, the concept of spatial confidence map has not been leveraged for multi-agent collaborative perception. The authors proposed an architecture to realize the concept and used the encoded information to send critical information for 3D object detection. The reviewer appreciates the proposed architecture as it improves interpretability. In addition, because of the design of a spatial confidence map, the framework can send sparse yet critical information to other agents and show improvement in 3D object detection.
3. Promising performance in 3D object detection AP and communication volume usage is demonstrated on three datasets, OPV2V, V2X-Sim, and CoPerception-UAVs. The experiments empirically demonstrate the effectiveness of the proposed framework.
4. Qualitative evaluations show that the proposed method **exhibits** promising explainability. Toward trustworthy multi-agent collaborative perception, such attributes would be much appreciated.


Weaknesses:
1. Related work: The reviewer found that the proposed spatial confidence map is highly relevant to the probabilistic occupancy map and the concept of a local grid map for robotics applications. The reviewer encourages the authors to acknowledge the existence of these works.
    1. H. Li, M. Tsukada, F. Nashashibi, and M. Parent, "Multivehicle Cooperative Local Mapping: A Methodology Based on Occupancy Grid Map Merging," in *IEEE Transactions on Intelligent Transportation Systems*, vol. 15, no. 5, pp. 2089-2100, Oct. 2014
    2. N. Deo and M. M. Trivedi, "Convolutional Social Pooling for Vehicle Trajectory Prediction," CVPR 2018
    3. C. Chen, Y. Liu, S. Kreiss, and A. Alahi, "Crowd-Robot Interaction: Crowd-aware Robot Navigation with Attention-based Deep Reinforcement Learning," *ICRA 2019*
2. Perfect localization: According to the reviewer's understanding, the proposed framework relies on the accurate localization of all agents in a scene. Without the assumption, a spatial confidence map cannot be applied easily. The reviewer felt that noisy localization could lead to significant performance degradation for 3D object detection. Toward real-world applications, it is inevitable to encounter such issues. It would be an informative study to analyze the sensitivity of the proposed method on noisy localization. Moreover, what is the corresponding performance compared with the existing STOA?
3. Communication volume: The reviewer has difficulty understanding how the authors adjust the communication volume shown in Figures 3 and 4. The reviewer's best guess is relevant to Figure 2, shown in the supplementary material. However, it is hard to reproduce the results without a clear explanation.
4. Ablative studies:
    1. What is the baseline implementation without using SCM? Is it simply using BEV features?
    2. Spatial confidence-aware message fusion module: What if we use max-pooling/concatenation for fusion?
5. Can the proposed algorithm solve pedestrian or other traffic participant detection? What could be the potential limitations?

---

> ### Author Response · Authors · 2022-08-01
> **Address Reviewer VaH1's [Weakness 4-5]**
>
> ### [Weakness 4a:] What is the baseline implementation without using SCM? Is it simply using BEV features?
>
> **1)** The baseline implementation without SCM means the attention weight used to fuse the BEV features in equation (3) $\mathbf{W}\_{j\rightarrow i}^{(k)} = {\rm MHA}\_{\rm W}\left(\mathcal{F}\_{i}^{(k)}, \mathcal{Z}\_{j\rightarrow i}^{(k)}, \mathcal{Z}\_{j\rightarrow i}^{(k)}\right) \odot  \mathbf{C}^{(k)}\_j \in \mathbb{R}^{H\times W}$ is decided only by the multi-head attention, given by $\mathbf{W}\_{j\rightarrow i}^{(k)} = {\rm MHA}\_{\rm W}\left(\mathcal{F}\_{i}^{(k)}, \mathcal{Z}\_{j\rightarrow i}^{(k)}, \mathcal{Z}\_{j\rightarrow i}^{(k)}\right)\in \mathbb{R}^{H\times W}.$
>
> **2)** The comparison with/without SCM is to validate that the spatial confidence map explicitly reflecting the perceptually critical level can provide useful prior for attention learning.
>
> **3)** The baseline implementation without using SCM uses the augmented BEV features that aggregate the received messages from the other agents.
>
> ### [Weakness 4b:] What about max-pooling/concatenation for message fusion?
>
> **1)** Previous works have validated that SOTA methods, such as When2com and DiscoNet,V2V, are superior to these two simple fusion baselines, so we did not report these two results in the paper.
>
> **2)** We further compare *Where2comm* with max-pooling/concatenation-based on all three datasets, results are shown as followed.
>
> |Fusion|CoPerception-UAVs|OPV2V|V2X-Sim|
> |-|-|-|-|
> |No Collaboration|57.84|22.66|45.80|
> |Max-Pooling|58.93|35.17|56.80|
> |Concatenation|59.00|25.68|56.50|
> |When2com|61.63|19.53|46.80|
> |V2VNet|59.82|37.47|55.30|
> |DiscoNet|59.74|36.00|58.00|
> |Where2comm|**64.83**|**47.30**|**59.10**|
>
> We see that:
> 1) Max-pooling and Concatenation outperform No Collaboration; while they are inferior to *Where2comm*, implying that with a simple fusion module, collaborative perception can improve single-agent perception, while the improvement is limited;
> 2) *Where2comm* significantly outperforms Max-pooling and Concatenation, improving about 10\% on CoPerception-UAVs, 30\% on OPV2V, 4\% on V2X-Sim, implying that with a proper fusion module, collaboration can significantly boost the single-agent perception.
>
> ### [Weakness 5] Solve pedestrian or other traffic participant detection? Potential limitations?
> **1)** Yes, of course. Our core idea is to enable a spatial confidence map for each agent, where each element reflects the perceptually critical level of a corresponding spatial area. Based on this idea, the proposed *Where2comm* can be easily generalized to many other tasks by substituting the detection head, loss function, and supervision;
>
> **2)** The potential limitation is that the feature maps need to be upgraded with higher resolution to adapt to the small object size, causing larger computational costs.
>
> **3)**  This question also inspires us to expand experiments to small objects in the future. We believe that collaborative perception might be more meaningful for small object detection than large object detection, because small object detection is more challenging, and a little more perceptual information via collaboration might bring significant benefits.

---

> > ### Comment · Reviewer_VaH1 · 2022-08-08
> > **Responses to the authors' feedback**
> >
> > Thanks for the authors’ time and efforts in responding to the questions.
> > 1. Please do cite relevant papers in the final version if the paper is accepted.
> > 2. Thanks for conducting the sensitivity analysis. Please incorporate the results and provide detailed discussions. The results show that the proposed framework is at least as robust to the localization error compared to existing methods. However, the reviewer still suspects that the proposed method would degrade significantly when localization errors increase (e.g., 1m or more), which would be observed when buildings surround vehicles. While the work focuses on the communication volume, the reviewer believes the spatial confidence map plays a vital role in the goal. Therefore, the reviewer suggests the authors conduct further sensitivity analysis and highlight the potential limitation deployed in the real world in the limitation section.
> > 3. Thanks for the clarification. Please incorporate the discussion in the final version if the paper is accepted. In addition, the source code for the calculation should be released as promised by the authors.
> > 4 (a). Thanks for the clarification.
> > 4.(b). Thanks for the additional experiment, and please do present the additional and highlight the importance of the fusion module.
> > 5. The reviewer agrees that demonstrating the effectiveness of the proposed method on small objects (pedestrians and bicyclists) is essential. In particular, these objects are small in BEV and could be hard to detect in a spatial confidence map. The reviewer encourages the authors to study the challenging setting!

---

> > > ### Author Response · Authors · 2022-08-09
> > > **Response to Reviewer VaH1's [feedback 4]**
> > >
> > > 4. Thanks for your suggestion on the small object detection setting! This is indeed a very promising future direction!
> > >
> > > **i)** So far, collaborative 3D detection of small objects has not been well benchmarked. Previous collaborative perception works only focus on vehicles and only the results of vehicle detection are reported. The OPV2V dataset and the provided parsed V2X-Sim dataset only contain vehicle category. **We will try to fill this gap and provide the first collaborative 3D detection benchmark for small objects in the final version, if the submission is accepted**;
> > >
> > > **ii)** We expect that the overall perception performance may decrease, while the gain of our proposed collaborative perception method may increase. Since small objects are more prone to occlusion problems and fail in single-agent views, obtaining a little more perceptual information through collaboration may bring significant benefits;
> > >
> > > **iii)** In addition, to address small object detection in spatial confidence maps without adding too much communication cost, we can include an adaptive pyramid feature map resolution selection strategy to flexibly select the most effective and efficient collaborative messages for variable-sized objects, i.e., choose a higher resolution for small objects and a relatively lower resolution for larger objects.

---

> > > > ### Comment · Reviewer_VaH1 · 2022-08-09
> > > > **Thank you for the responses**
> > > >
> > > > Thank you for your time and effort in this response. The additional analysis is helpful, and detailed explanations are convincing. Please incorporate the discussions in the final version if the paper is accepted. All my significant concerns have been addressed. Thank you so much for the extraordinary efforts and excellent work.

---

> > > ### Author Response · Authors · 2022-08-09
> > > **Response to Reviewer VaH1's [feedback 1-3]**
> > >
> > > Thanks very much for reading our responses and providing your feedback! As we promised, if the submission gets accepted, in the final version, we will definitely cite the relevant papers, release the codes, and highlight the mentioned additional experiments, sensitivity analysis, and detailed limitation discussions. Here are our detailed responses.
> > >
> > > 1. We will definitely cite those papers related to the probabilistic occupancy map in the final version, if the submission is accepted.
> > >
> > > 2. **i)** We will definitely incorporate the sensitivity analysis results and provide detailed discussions in the main paper.
> > >     **ii)** For the sensitivity analysis about localization errors, we want to further emphasize that:
> > >     * In real-world scenarios, the localization issue has been widely addressed, achieving an error within 0.3m[1,2,3]; following this realistic setting, the previous SOTA V2X-ViT and V2VNet evaluate the localization robustness with noise no more than 0.5m, so do we, and our method achieves promising performance under this realistic setting;
> > >     * We further evaluate extremely large anomaly localization errors up to 1.5m. Results are shown in the below tables. We see that: **1)** **Where2comm** degrades smoothly when localization error increases and consistently outperforms No Collaboration, V2VNet, and DiscoNet, even when the localization error increases to 1m, reflecting the robustness; **2)** **Where2comm** will encounter performance limitations when the localization error exceeds 1.1m on CoPerception-UAV, 1.5m on V2X-Sim, and it drops by about 2% at 1.5m, which is an extremely abnormal case in the real world. In this case, a slight decrease in perception may not be a bottleneck, because the planning and control modules in the autonomous system may encounter severe collision risks;
> > >     * In addition, **where2comm** can easily extend to a deformable transformer architecture to further introduce collaborative feature fusion flexibility, and alleviate the effect of the localization error.
> > >
> > >     **iii)** For other potential limitations deployed in the real world, we have appended a section of **discussion on the realistic limitations** in the revised supplementary material including realistic communication latency, time synchronization, noisy localization, attack, and data availability. We will highlight them in the limitation session in the main paper in the final version if the submission is accepted.
> > >
> > > ```[CoPerception-UAVs]```
> > > |Std(m)|0|0.1|0.2|0.3|0.4|0.5|0.6|0.7|0.8|0.9|1.0|1.1|1.2|1.3|1.4|1.5|
> > > |-|-|-|-|-|-|-|-|-|-|-|-|-|-|-|-|-|
> > > |No Collaboration|57.84|57.84|57.84|57.84|57.84|57.84|57.84|57.84|57.84|57.84|57.84|57.84|57.84|57.84|57.84|57.84|57.84|
> > > |V2VNet|59.82|59.42|58.85|57.89|56.41|54.87|53.68|52.25|51.49|50.21|49.90|49.39|48.41|48.17|47.49|47.21|
> > > |DiscoNet|59.74|59.61|59.51|58.88|58.06|57.02|56.35|55.61|54.46|54.00|53.82|53.26|52.64|52.56|52.25|51.89|
> > > |Where2comm|64.83|64.71|64.36|63.98|63.22|62.07|61.27|60.19|59.01|58.51|58.24|57.56|56.87|56.46|56.40|55.93|
> > >
> > > ```[OPV2V]```
> > > |Std(m)|0|0.1|0.2|0.3|0.4|0.5|0.6|0.7|0.8|0.9|1.0|1.1|1.2|1.3|1.4|1.5|
> > > |-|-|-|-|-|-|-|-|-|-|-|-|-|-|-|-|-|
> > > |No Collaboration|22.66|22.66|22.66|22.66|22.66|22.66|22.66|22.66|22.66|22.66|22.66|22.66|22.66|22.66|22.66|22.66|
> > > |V2VNet|37.47|37.24|35.85|33.08|30.52|28.26|25.79|24.68|23.40|22.58|21.59|21.17|20.53|20.09|19.77|19.43|
> > > |DiscoNet|36.00|35.69|35.56|33.16|30.76|29.49|27.46|26.32|24.96|23.69|22.64|21.94|21.34|21.12|20.93|20.79|
> > > |Where2comm|47.30|46.31|45.45|44.13|41.83|40.08|38.30|36.45|34.85|33.73|32.81|32.22|31.75|31.00|30.81|30.49|
> > >
> > >
> > >
> > > ```[V2X-Sim]```
> > > |Std(m)|0|0.1|0.2|0.3|0.4|0.5|0.6|0.7|0.8|0.9|1.0|1.1|1.2|1.3|1.4|1.5|
> > > |-|-|-|-|-|-|-|-|-|-|-|-|-|-|-|-|-|
> > > |No Collaboration|45.8|45.8|45.8|45.8|45.8|45.8|45.8|45.8|45.8|45.8|45.8|45.8|45.8|45.8|45.8|45.8|
> > > |V2VNet|55.3|54.6|54.0|52.1|51.0|50.7|50.3|50.0|49.5|48.8|48.4|47.9|47.2|46.0|44.6|43.3|
> > > |DiscoNet|58.0|57.9|57.7|57.5|57.3|56.8|56.0|55.4|54.6|53.6|52.5|51.1|48.9|47.2|45.8|43.6|
> > > |Where2comm|59.1|58.8|57.9|57.7|57.5|57.3|56.8|56.7|56.0|54.9|53.3|51.4|49.7|47.8|45.9|43.9|
> > >
> > > 3. We will definitely release the related codes as soon as possible if the paper is accepted, and incorporate the additional experiments, sensitivity analysis, and detailed discussions in the final version.
> > >
> > >
> > > [1] Elbaz, Gil et al. “3D Point Cloud Registration for Localization Using a Deep Neural Network Auto-Encoder.” 2017 IEEE Conference on Computer Vision and Pattern Recognition (CVPR) (2017): 2472-2481.
> > >
> > > [2] Wang, Wei et al. “DeepPCO: End-to-End Point Cloud Odometry through Deep Parallel Neural Network.” 2019 IEEE/RSJ International Conference on Intelligent Robots and Systems (IROS) (2019): 3248-3254.
> > >
> > > [3] Yuan, Yunshuang and Monika Sester. “Leveraging Dynamic Objects for Relative Localization Correction in a Connected Autonomous Vehicle Network.” ArXiv abs/2205.09418 (2022): n. pag.

---

> ### Author Response · Authors · 2022-08-01
> **Address Reviewer VaH1's [Question about localization noise] and [Weakness 1-3]**
>
> ### [Weakness1:] Acknowledge occupancy map-related works?
> Thanks for pointing out the related works, we will cite them in the revised version.
>
> ### [Question & Weakness2:] The sensitivity of the proposed method & existing SOTA on noisy localization?
> **1)** **Where2comm is at least as robust to the localization noise as previous SOTAs.** We follow the localization noise setting in V2VNet and V2X-ViT (Gaussian noise with a mean of 0m and a standard deviation of 0m-0.6m) and assess the robustness against realistic localization noise on all three datasets. Results are shown in the below tables; **also see Fig.7 in the revised supplementary material.**
>
> ```[CoPerception-UAVs]```
> |Std(m)|0|0.1|0.2|0.3|0.4|0.5|0.6|
> |-|-|-|-|-|-|-|-|
> |No Collaboration|57.84|57.84|57.84|57.84|57.84|57.84|57.84|
> |When2com|61.63|61.63|61.54|61.14|60.74|60.23|59.81|
> |V2VNet|59.82|59.42|58.85|57.89|56.41|54.87|53.68|
> |DiscoNet|59.74|59.61|59.51|58.88|58.06|57.02|56.35|
> |Where2comm|64.83|64.71|64.36|63.98|63.22|62.07|61.27|
>
> ```[OPV2V]```
> |Std(m)|0|0.1|0.2|0.3|0.4|0.5|0.6|
> |-|-|-|-|-|-|-|-|
> |No Collaboration|22.66|22.66|22.66|22.66|22.66|22.66|22.66|
> |When2com|19.53|19.35|18.78|18.04|17.07|16.30|15.26|
> |V2VNet|37.47|37.24|35.87|33.08|30.52|28.26|25.79|
> |DiscoNet|36.00|35.69|35.56|33.16|30.76|29.49|27.46|
> |Where2comm|47.30 |46.31| 45.45 |44.13| 41.83| 40.08| 38.30|
>
> ```[V2X-Sim]```
> |Std(m)|0|0.1|0.2|0.3|0.4|0.5|0.6|
> |-|-|-|-|-|-|-|-|
> |No Collaboration|45.8|45.8|45.8|45.8|45.8|45.8|45.8|
> |When2com|46.8|46.8|46.7|46.4|46.2|46.1|45.7|
> |V2VNet|55.3|54.6|54.0|52.1|51.0|50.7|50.3|
> |DiscoNet|58.0|57.9|57.7|57.5|57.3|56.8|56.0|
> |Where2comm|59.1|58.8|57.9|57.7|57.5|57.3|56.8|
>
> We see that:
> 1) Overall the collaborative perception performance degrades with the increasing localization noise, while ***where2comm* outperforms previous SOTAs under all the localization noise levels.**
> 2) *Where2comm* keeps outperforming *No Collaboration* while V2VNet/DiscoNet fails when noise is over 0.4m/0.5m on CoPerception-UAVs. The reasons are i) the powerful transformer-based fusion module attentively selects the most perceptually feasible collaborative feature; ii) the spatial confidence map hints help to filter out perceptually infeasible noisy features, mitigating noise localization distortion effects.
>
> **2)** We want to emphasize that our main focus is the communication bandwidth and perception performance trade-off, as previous SOTAs (DiscoNet,When2com). It is a critical problem in the current collaborative perception system, decoupled with the noisy localization problem. We did not make any specific design for handling localization noise issue. Meanwhile, the proposed system can integrate many methods that are specifically designed to address the noisy localization issue, such as [1] published at CoRL20.
>
> ### [Weakness3:] Calculation of communication volume?
> **1)** Sorry for the confusion! Communication volume is relevant to Figure 2 in the supplementary material and mentioned in Line 256 in the paper. It counts the message size by byte in log scale with base $2$.
>
> **2)** Mathematically, for the selected sparse feature map $\mathcal{Z}_{i\rightarrow j}^{(k)}=\mathbf{M}\_{i\rightarrow j}^{(k)}\odot\mathcal{F}_i^{(k)} \in \mathbb{R}^{H\times W \times D}$, the communication volume is $\text{log}_2\left(|\mathbf{M}\_{i\rightarrow j}^{(k)}| \times D \times 32 / 8\right)$, where $|\cdot|$ denotes the L0 norm counting the non-zero elements in the binary selection matrix, this is, counting the total spatial girds need to be transmitted, and $D$ denotes the channel dimension, $32$ is multiplied as float32 data type is used, $8$ is divided as the metric byte is used.
>
> **3)** Our communication volume is the same as DiscoNet, the only difference is that our log base is $2$, while it is $10$, so our number is about $3.32$ times theirs. We will release all the source codes, including communication volume calculation.
>
> **We have updated the supplementary material to reflect this.**
>
> [1] Vadivelu, Nicholas et al. “Learning to Communicate and Correct Pose Errors.” CoRL (2020).

---

### Official Review · Reviewer_cKVM · 2022-07-12

**Rating:** 6
**Confidence:** 3
**Soundness:** 3 good
**Presentation:** 3 good
**Contribution:** 3 good

**Summary:**

The paper presents a collaborative perception model that specifically focuses on reducing communication using spatial confidence maps. These maps allow agents to decide which areas of the scene they need further information from, as well as which areas of the scene they have relevant information. The authors use attention to allow the agents to attend their, and other, confidence maps to create a communication graph. The authors then use a Transformer Architecture with Multi-Head Attention to fuse the features from multiple agents. The fused features will act as the agents input features in the next iteration, and they are also fed to a decoder that can regress the relevant information (class and bounding box). Overall, the approach achieves significantly lower communication rates, while enabling the network to also use similar communication Volumes as the state of the art.


**Questions:**

1. You mention using distance between the sensor and the feature to perform positional encoding. How does this work for a monocular camera? Do you assume known depth?
2. Can you elaborate in the warping function from iage to BEV? How is this performed? This is an active area of research, see e.g. Saha, Avishkar, et al. "Translating images into maps." arXiv preprint arXiv:2110.00966 (2021). and Roddick, Thomas, and Roberto Cipolla. "Predicting semantic map representations from images using pyramid occupancy networks." Proceedings of the IEEE/CVF Conference on Computer Vision and Pattern Recognition. 2020.
3. Why BEV? Have you looked at other ways of unifying the data? Could this be a learned transformation applied to the data?
4. For BEV, how robust is your system to distortions introduced by ground projection?

**Limitations:**

The work is only evaluated in simulation. This presents important challengs that must be solved. The authors should include limitations in terms of time alignment, assumptions about sensor modalities (is depth required?) and other consequences of moving into the real world. The authors mention expanding their idea into the temporal dimension, which seems like a very good idea. Although another possible limitation of their approach is that as the dimensionality of the problem gets bigger, the attention component might not be capable of attending to all the relevant information, resulting in sub-standard communication.

**Strengths And Weaknesses:**

The paper is well presented, generally reads well and provides a good case for the authors Spatial Confidence maps as a way to reduce communication and increase performance in collaborative perception. The quantitative data suggest the model can outperform the state of the art on the evaluated datasets. It shows a strong ability to reduce the amount of data, while still outperforming all the benchmarks. There is a massive reduction in the amount of data used, which should be commended. The qualitative data shows much improved performance in the detection. Furthermore, the ablation study cleary shows improvements introduced by each part of the architecture.

In terms of weaknesses, the main limitation is that the models is only evaluated on synthetic datasets with favourable conditions. A discussion of limitations including things like time synchronisation, noise, availability of data and other considerations are important to present on an approach that relies on collaborative perception. Additionally, the paper has various spelling and grammar mistakes that should be rectified.

---

> ### Author Response · Authors · 2022-08-01
> **Response to Reviewer cKVM's comments on [Question 4-5]**
>
> ### [Q4] For BEV, robustness to distortions introduced by ground projection?
> **1)** As the answer to [Q2] stated, for monocular input, instead of directly projecting 2D features to flat ground space, we first lift them to 3D voxel space and then flatten them to the ground. This design considers all the possible depths/altitudes, introducing flexibilities in the projection, and mitigating the distortion effect. However, distortion is unavoidable due to information loss in imaging.
>
> **2)** *Where2comm* achieves consistent improvements on three datasets, CoPerception-UAVs (6.62\%), OPV2V (25.81\%), V2X-Sim (1.9\%), demonstrating robustness.
>
> **3)** The reasons for *Where2comm* being robust to distortions are:
> * Collaboration compensates the information loss in the distorted features in the single-agent view, as the distorted features for one agent may be clearly perceived by other agents;
>
> * The fusion module specifically uses i) powerful transformer architecture to attentively select the more perceptually feasible features over the distorted features, and ii) spatial confidence map to filter out perceptually infeasible distorted features.
>
> **4)** *Where2comm* can easily extend to a deformable transformer architecture to further alleviate the distortions.
>
> ### [Q5] Grammar issues?
> Thanks for pointing it out! We will modify those errors carefully.

---

> > ### Comment · Reviewer_cKVM · 2022-08-09
> > **Response to Rebuttal**
> >
> > Hi,
> >
> >  Thank you for taking the time to address my concerns in your rebuttal.
> >
> > I understand it is challenging to assess these contributions outside of simulation, and do appreciate the table demonstrating your evaluation is more thorough. It makes the work much clearer when I look at it in this context. I am also happy to see there is scope to include some real datasets in the approach and I think this would significantly increase the impact of the publication. I am also glad to see some analysis of realistic limitations both in your response and the supplemenatary material. Q1-4 have also been answered clearly, although it would be relevant to include information about the CaDDN protocol followed in section 4.1 where this is discussed.
> >
> > Ovearll, I am happy that my concerns have been addressed and that the publication now contains more information about the implementation details and discussions of realistic limiations to this approach.

---

> > > ### Author Response · Authors · 2022-08-10
> > > **Response to Reviewer cKVM's Feedback**
> > >
> > > Thanks very much for reading our responses! In the final version, we will benchmark our method and previous SOTAs on this just-released real dataset, DAIR-V2X, incorporate more details, discussions and highlight the realistic limitations, if the submission is accepted.
> > >
> > > Particularly, we appreciate your interest in our BEV representation and transformation. Here are some further responses:
> > >
> > > 1. We will incorporate more details and discussions about representation and transformation in Section 4.1;
> > >
> > > 2. The proposed Where2comm is friendly to many other unified spaces and warping choices. We did not specifically design the way to unify data and the transformation to warp data, as our main focus is to promote the communication cost and perception performance trade-off;
> > >
> > > 3. This is indeed a very promising future direction to further enhance the collaborative space and apply more effective transformation methods to further improve the perception performance of Where2comm.

---

> ### Author Response · Authors · 2022-08-01
> **Response to Reviewer cKVM's comments on [Question 1-3]**
>
> ### [Q1:] Positional encoding for camera? Assume known depth?
> For the monocular camera, **the depth is unknown** and we need to **estimate** the depth. The steps are:
> 1) CaDDN[1] (CVPR 2021 Oral) is applied to estimate the depth distribution for each image feature point.
> 2) Each feature point is wrapped from the 2D image space to the 3D physic space according to the known camera parameters.
> 3) The 3D voxel features are flattened to BEV features. For each BEV feature point, the sensor positional encoding is conditioned on the distance between the known sensor coordinates and each BEV gird's coordinate in the 3D physic space.
>
> ### [Q2:] Elaborate in the warping function from image to BEV.
> 1) We follow CaDDN[1], which is a recent and effective method to warp image feature to BEV feature. As it is not our novel contribution, the details are not unfolded in our paper.
> 2) The main steps are unfolded in the answer to [Q1]. Briefly, for each image feature point locates at $(u,v)$, given the estimated categorical depth $d_i$, and the known camera projection matrix $\mathbf{P}\in\mathbb{R}^{3\times4}$, 3D physical space coordinates $[x,y,z]^T$ is calculated conditioned on the image feature coordinates $[u,v,d_i]^T$ based on the projection function:  $[u,v,d_i]^T=\mathbf{P}\cdot[x,y,z,1]^T$. And more details could refer to~\cite{CaDDN}, and also we will release the related code.
> 3) Thanks a lot for pointing out the related warping methods, we will cite them in the revised version.
>
> ### [Q3] Why BEV? Other ways of unifying the data? Learnable transformation?
> 1) A BEV representation provides a common spatial coordinate system to unify the spatial-temporal data from multiple agents and multiple modalities. It also nicely preserves both geometric and semantic information and is suitable for various tasks, such as perception, prediction, and planning. Additionally, BEV-based techniques are an emerging research topic and have achieved outstanding performances on both camera-only and LiDAR-based 3D detection[2,3,4]. Furthermore, previous collaborative perception works, including V2VNet, DiscoNet, and OPV2V, all adopt the BEV representation.
> 2) Besides the BEV representation, data could also be unified in other spaces, such as the 2D image space, raw 3D spatial space, and 3D voxel space. Each one has its advantages and disadvantages. In the 2D image space, severe geometric distortion is introduced, which makes it less effective for geometric-oriented tasks, such as 3D object recognition. In the 3D spatial space, for monocular inputs, an offline depth estimation network is required to lift the 2D image into pseudo-3D point clouds, this would cause accumulated errors and semantic loss. In the 3D voxel space, geometric structure and semantic clues can be preserved, while a prohibitive efficiency bottleneck is encountered as the computation and communication costs increase exponentially.
> 3) In this work, we apply the fixed geometric transformation to warp image feature to BEV, following the powerful CaDDN[1]. Meanwhile, some recent methods, such as BEVFormer[2], BevFusion[3], and PETR[4] show that the learnable transformation can achieve great performance.
>
> Additionally, we want to emphasize that our focus is the collaboration strategy to promote the communication cost and perception performance trade-off. So we did not specifically design the way to unify data and the transformation to warp data. Our core idea is to enable a spatial confidence map for each agent, where each element reflects the perceptually critical level of a corresponding spatial area. Based on this idea, **the proposed Where2comm is friendly to many other unified spaces and warping choices**.
>
> [1] Reading, Cody et al. “Categorical Depth Distribution Network for Monocular 3D Object Detection.” CVPR (2021): 8551-8560.
>
> [2] Li, Zhiqi et al. “BEVFormer: Learning Bird's-Eye-View Representation from Multi-Camera Images via Spatiotemporal Transformers.” ArXiv abs/2203.17270 (2022): n. pag.
>
> [3] Liu, Zhijian et al. “BEVFusion: Multi-Task Multi-Sensor Fusion with Unified Bird's-Eye View Representation.” ArXiv abs/2205.13542 (2022): n. pag.
>
> [4] Liu, Ying-Hao et al. “PETR: Position Embedding Transformation for Multi-View 3D Object Detection.” ArXiv abs/2203.05625 (2022): n. pag.

---

> ### Author Response · Authors · 2022-08-02
> **Response to Reviewer cKVM's comments on [Main Concern]**
>
>
> ### Only simulation datasets?
> **1)** By the submission deadline, there was no publicly available real-world dataset for collaborative perception. **Only two simulation datasets (V2X-Sim and OPV2V) were available** and we have done extensive experiments on both datasets.
>
> **2)** Compared with related previous works, this work presents much more comprehensive experimental results. As shown below table, previous works only reported their results on a single dataset with a single view and modality, while Where2comm is validated on three datasets, including all the released 3D collaborative perception benchmarks (V2X-Sim and OPV2V) and a self-organized dataset (CoPerception-UAVs), covering diverse views and modalities.
>
> |Method|Venue|Dataset|Released|Real/simulation|View|Perception|Modality|
> |-|-|-|-|-|-|-|-|
> |V2VNet|ECCV20|V2V-Sim|X|simulation|Vehicle|3D Det.|PointCloud|
> |Who2com|ICRA20|Airsim-Map|&#10004;|simulation|Drone|2D Seg.|Image|
> |When2com|CVPR20|Airsim-Map|&#10004;|simulation|Drone|2D Seg.|Image|
> |AAOMAC|ICCV2021|V2V-Sim|X|simulation|Vehicle|3D Det.|PointCloud|
> |DiscoNet|Neurips21|V2X-Sim|&#10004;|simulation|Vehicle|3D Det.|PointCloud|
> |OPV2V|ICRA22|OPV2V|&#10004;|simulation|Vehicle|3D Det.|PointCloud|
> |V2X-ViT|ECCV22|OPV2V|&#10004;|simulation|Vehicle|3D Det.|PointCloud|
> |SyncNet|ECCV22|V2X-Sim|&#10004;|simulation|Vehicle|3D Det.|PointCloud|
> |Where2comm|Submit to Neurips22|V2X-Sim,OPV2V,CoPerception-UAVs|&#10004;|simulation|Vehicle & Drone|3D Det.|Image&PointCloud|
>
> **3)** Organizing real-world datasets is extremely expensive and laborious, so that most previous works are evaluated on simulated datasets. We are delighted to see that Dair-V2X, a real recorded collaborative perception dataset, was accepted to CVPR 2022 (**June** 2022). However, the data and code were not fully released and cannot be used at the submission deadline. We also noticed that Dair-V2X have been fully released around two weeks ago (mid July). So we will be able to include the results on Dair-V2X in the revised version.
>
> ### Limitations in moving to real world?
> **1)** We agree with the reviewer that there is a lot of real-world challenges in a collaborative perception system, such as ideal communication latency and localization error. However, a single paper cannot and should not fully address all of them. Specifically, we want to emphasize the following two points:
> * Solutions to each real-world challenge could be published individually. For example, AAOMAC published at ICCV21 specifically tackles the attack issue in collaborative perception; and SyncNet published at ECCV22 specifically tackles the latency issue in the collaborative perception system.
> * In this work, we focus on the biggest real-world challenge in current collaborative perception systems; that is, the trade-off between communication bandwidth and perception performance. This challenge has been actively addressed in previous works (Who2com,When2com,V2vnet,DiscoNet). Because collaborative perception is enabled and also severely limited by the communication capacity, which is critically reflected in the highly dynamic and limited bandwidth in real-world communication systems. Where2comm flexibly adapts to various communication bandwidths, achieving superior performance-bandwidth trade-off.
>
> **2)** Here we discuss some realistic limitations; **also see the revised supplementary material.**
> * For other realistic communication issues such as **latency**, *where2comm* communicates strategically when necessary, rather than all the time or everywhere, to reduce the possibility of encountering communication problems. In addition, a prediction module could be integrated to estimate the missed or delayed frames according to the historically received frames. And by focusing on the informative spatial regions, *where2comm* can reduce the estimation difficulty.
> * For the **time synchronization** issue, by using the powerful transformer architecture-based fusion module, *where2comm* can attentively augment the features with the received asynchronous features from other agents. In addition, *where2comm* can introduce positional encoding conditioned on delay time and easily extend to global multi-head attention to further reduce the effects of time synchronization.
> * For the **noisy localization** issue, *where2comm* exchanges the intermediate features among agents, which has a relatively low spatial resolution, thus is relatively robust to noisy pose. In addition, *where2comm* can easily extend to a deformable transformer architecture to further alleviate the feature distortion caused by the noisy localization.
> * For the **attack** issue, by focusing on specific spatial regions and attentively fusing the received features from other agents, *where2comm* is relatively less likely to be attacked.
> * For the **data availability**, *where2comm* works on both RGB and point cloud modalities, and is sensor-friendly, so it can be deployed on cheap camera sensors and lidar sensors.

---

### Meta-Review · Area_Chair_jrNK · 2022-08-20

**Recommendation:** Accept
**Confidence:** Certain

**Metareview:**

This paper proposes a multi-agent collaborative perception algorithm where agents exchange perceived sensors (e.g., LIDAR) and share their observations with other agents sparsily by maintaining spatial confidence maps that determine the communication connectivity matrix. Communication happens over multiple rounds and incoming messages are fused using multi-head attention. The method is evaluated on synthetic drone and car data from popular simulators, where it achieves superior results with significantly lower communication volume.

Reviewers praised the experiements and the performance of the model (cKVM, VaH1, Jgi5), the large reduction in communication overhead (cKVM, YUZ9), the novelty of the confidences-map based communication framework (VaH1, YUZ9, Jgi5).

Reviewer cKVM noted that the model was evaluated only on synthetic data (cKVM) and a missing section on data synchronisation and availability; in the rebuttal, authors argued about limited availability of simulated and real-world data, and explained how depth was extracted from RGB observations. Reviewer VaH1 suggested existing literature on local grid map pooling of probabilistic occupancy maps, questioned performance in case of noisy observations (the authors replied with additional results in those conditions), and doing ablation experiments. Reviewer YUZ9 had some questions, answered in the rebuttal. Reviewer Jgi5 had several questions on the complexity of the model, the fairness of some comparisons and the reproducibility of the method - points addressed in the rebuttal.

The reviewers agree on the score (6) and on the fact that the paper should be accepted, and the AC concurs.

Sincerely,
Area Chair


**Award:**

No

---

### Decision · Program_Chairs · 2022-09-14

Accept